# Phylogenetic Partitioning of Gansu Flora: Unveiling the Core Transitional Zone of Chinese Flora

**DOI:** 10.3390/plants12173060

**Published:** 2023-08-25

**Authors:** Zizhen Li, Qing Tian, Peifang Chong, Weibo Du, Jia Wei, Rong Huang

**Affiliations:** 1College of Forestry, Gansu Agricultural University, Lanzhou 730070, China; lizizhen1206@163.com (Z.L.); dsjzxw1@163.com (W.D.); huangr@st.gsau.edu.cn (R.H.); 2Jinchang Municipal People’s Government, Jinchang 737100, China; 3Research Institute of Forestry, Chinese Academy of Forestry, Beijing 100089, China; 15117290190@163.com; 4Lanzhou Institute of Landscape Gardening, Lanzhou 730070, China

**Keywords:** floristic regions, phylogenetic relationships, phylogenetic beta diversity, spatial turnover, seed plants

## Abstract

Floristic regions, conventionally established using species distribution patterns, have often overlooked the phylogenetic relationships among taxa. However, how phylogenetic relationships influence the historical interconnections within and among biogeographic regions remains inadequately understood. In this research, we compiled distribution data for seed plants in Gansu, a region of significant biogeographic diversity located in northwestern China.We proposed a novel framework for floristic regions within Gansu, integrating distribution data and phylogenetic relationships of genera-level native seed plants, aiming to explore the relationship between phylogenetic relatedness, taxonomic composition, and regional phylogenetic delineation. We found that (1) phylogenetic relatedness was strongly correlated with the taxonomic composition among floras in Gansu. (2) The southeastern Gansu region showed the lowest level of spatial turnover in both phylogenetic relationships and the taxonomic composition of floristic assemblages across the Gansu region. (3) Null model analyses indicated nonrandom phylogenetic structure across the region, where most areas showed higher phylogenetic turnover than expected given the underlying taxonomic composition between sites. (4) Our results demonstrated a consistent pattern across various regionalization schemes and highlighted the preference for employing the phylogenetic dissimilarity approach in biogeographical regionalization investigations. (5) Employing the phylogenetic dissimilarity approach, we identified nine distinct floristic regions in Gansu that are categorized into two broader geographical units, namely the northwest and southeast. (6) Based on the phylogenetic graphic regions of China across this area.

## 1. Introduction

Understanding biogeographical regions is vital for exploring species distributions, biogeographical analyses, and biodiversity conservation and for formulating effective species conservation programs [1,2,3]. A central aim of biogeographical regionalization is to classify groups of organisms into meaningful geographical units at different scales for a better understanding the patterns of biodiversity [4]. Traditionally, biogeography regionalization was typically accomplished via the use of qualitative methods frequently centered around the principle of taxonomic endemism [5]. Qualitative methods based on taxonomic dissimilarities at species, generic, and familial levels have been applied to the process of biogeographical regionalization [2,6,7]. Several biomes and biogeographical regionalization schemes have been proposed, such as the world’s flora [8,9,10], East Asian Plants [11], the flora of China [12,13], and the biogeographical framing of the Gansu area [14,15]. These schemes offered spatially explicit framework to investigate biodiversity patterns on a large scale [5]. Nonetheless, a notable constraint observed in preceding studies is their failure to incorporate a deliberate examination of the phylogenetic interconnections among species. This oversight has led to an inadequacy in portraying the multifaceted evolutionary chronicle of floristic territories, as well as a deficiency in bioclimatic and palaeobiogeographic contextual elucidation [16,17].

At present, the importance of robust systems for classifying biogeographic patterns has been emphasized, particularly with the proliferation of extensive species distribution databases [18,19,20,21,22,23]. The utilization of phylogenetic information has gained momentum in improving our understanding of floristic assembly from an evolutionary perspective in the fields of ecology, biogeography, and evolutionary processes. It serves as a valuable tool for elucidating various aspects, including the phylogenetic composition, spatial phylogenetic patterns, evolutionary origins and diversification, the phylogenetic regionalization of floral communities, and the application of phylogenetic information in the realm of biodiversity conservation [24]. For example, the phylogenetic structure of a regional plant assemblage refers to the arrangement and distribution of plant species within a specific geographic area based on their evolutionary relationships. Qian et al. (2013), Li et al. (2014), and Swenson and Umaña (2014) revealed that the phylogenetic structure of regional plant assemblages is determined by environmental conditions and biogeographical history, ranging from the latitudinal gradient and latitudinal diversity gradient to environmental heterogeneity [25,26,27]. Phylogenetic diversity (PD) refers to the measurement of the evolutionary relatedness and diversity of species within a biological community or ecosystem [28]. Phylogenetic endemism (PE) pertains to the concentration of unique or endemic lineages in a specific geographic region [29]. Relative phylogenetic diversity (RPD) is a standardized measure that compares the observed phylogenetic diversity of a particular site or community to the phylogenetic diversity expected under a null model, often accounting for factors such as species richness [30]. Relative phylogenetic endemism (RPE) is similar to RPD. RPE is a relative measure that assesses the concentration of phylogenetically unique species within a specific area compared to a null model [30]. Mishler et al. (2014), Thornhill et al. (2016), and Scherson et al. (2017) used PD, PE, RPD, and RPE to better evaluate the spatial patterns of diversity and endemism from an evolutionary perspective and to explore the potential evolutionary and ecological causes of significant concentrations of diversity and endemism [28,29,30]. Verboom et al. (2009), Linder (2014), Chen et al. (2018), and Du et al. (2022) investigated the origins and diversification of floristic assemblages and explored the patterns of distribution and the dispersal of selected plant lineages from a particular floristic region using phylogenies and detailed distribution data from selected clades or entire flora [31,32,33,34]. Li et al. (2015), Daru et al. (2016), Slik et al. (2018), and Ye et al. (2019) used phylogenetic relationships and distribution data to identify distinct floristic regions globally and within China [4,5,13,35]. Also, in recent years, the evolutionary processes that maintain and generate biodiversity have been recognized as important factors that need to be accounted for in conservation planning [36,37,38]. Preserving a site containing a few highly diverse, distantly related lineages may be more beneficial than preserving a site with a large number of closely related taxa [39,40,41]. These works highlight the importance of incorporating phylogenetic information to improve our understanding of floristic assembly from an evolutionary perspective [42].

The phylogenetic approach and traditional methods in biological research exhibit distinct methodologies and focus areas. Traditional methods rely on observable traits, historical taxonomic classifications, and characteristics, which include morphology, ecology, and behavior, to infer relationships, ecological interactions, and species identities [4]. The phylogenetic approach centers on analyzing genetic diversity and evolutionary history among organisms using molecular data, revealing shared ancestry and divergence times. Integrating these approaches can offer a more comprehensive understanding of biodiversity, blending genetic insights with traditional ecological and taxonomic perspectives. The quantitative methods of phylogenetic information, rather than qualitative methods that rely on experts’ experience and knowledge, are more objective and repeatable [5].

Phylogenetic trees are utilized to quantify the phylogenetic relatedness of taxa across different geographic regions, which quantifies the phylogenetic turnover (phylogenetic beta diversity) among them. Furthermore, phylogenetic trees serve as a means to detect phylogenetic turnover, facilitating the evaluation of the distinct contributions of evolutionary and ecological processes in shaping current biodiversity patterns across extensive spatial extents [43]. Quantifying the spatial turnover of species or phylogenetic composition among sites is an essential step for identifying floristic regions [44]. Phylogenetic beta diversity reflects both ecological and evolutionary processes that shape biodiversity patterns [43], and it is a metric that quantifies the spatial variation in phylogenetic relatedness among species and is akin to taxonomic beta diversity, which measures the variation in species composition across space.

Gansu province is characterized by a complex physical geography, a rich biodiversity, and an ancient floristic component with many endemic taxa [45]. It encompasses four plant subregions: Holarctic flora, East Asian flora, Tethyan flora, and Qinghai–Tibet Plateau flora [5]; moreover, it has seven geomorphic types: mountains, plains, basins, plateaus, valleys, deserts, and the gobi desert [46]. Due to its unique characteristics, Gansu province serves as a pivotal intersection and transitional zone for the geographical distribution of numerous plant communities not only within China but also across Asia [47,48]. The region showcases a rich variety of vegetation types, with each adapted to its specific ecological niche. From lush alpine meadows to arid desert oases, Gansu’s plant communities span a wide range of altitudes, climates, and soil conditions. This diversity is shaped by the complex interplay of geographical features, including the Qilian Mountains, the Hexi Corridor, and the Yellow River. These various plant communities play a crucial role in maintaining local and regional biodiversity. Early studies on the flora of Gansu have been predominantly limited to taxonomic approaches, focusing on particular plant taxa or geographical regions [49,50,51]. And previous studies in plant phylogenetics have predominantly concentrated on geographically homogeneous hotspots, such as North America [25], Southern Africa [35], the Lesser Antillean of Central America [27], and Yunnan of China [4].

However, exploring the differences in plant diversity in the evolutionary dimension and understanding the distribution boundaries of different plant regions in Gansu with respect to complex geographical environments, diverse climates, and significant differences in vegetation is extremely attractive. No prior investigation has integrated both phylogenetic and spatial aspects to explore the evolutionary trajectory of the Gansu flora. We lack an understanding of the critical evolutionary dimension of floristic diversity in this area. Investigating the phytogeographic regionalization of this area is of great significance and value in enhancing our understanding of the differentiation and historical progression in transition zones of Chinese and even Asian floras.

Phytogeographic regionalization holds crucial significance in comprehending the evolutionary dimension of floristic diversity, particularly in the specified area [1]. By investigating the distribution patterns and evolutionary histories of plant species, it offers insights into conservation strategies, biogeographic evolution, ecosystem functioning, and taxonomic research [52]. To reinforce regionalization, integrating data from various species groups helps unveil co-evolutionary relationships, predict ecosystem shifts, and enhance our understanding of the region’s history and potential future changes [53]. This approach not only aids in delineating phylogenetically informed floristic regions but also deepens our grasp of phytogeographical establishment and identification [54].

In this study, we investigated the relationship between taxonomic and phylogenetic beta diversity, with a particular focus on Gansu floristic assemblages. To achieve this, we analyzed the spatial turnover patterns of Gansu floristic assemblages from both taxonomic and phylogenetic perspectives, employing a genus-level phylogeny of Gansu seed plants along with county-level distribution data. Our objective is to provide valuable insights into the demarcation of phylogenetically informed floristic regions and to enhance the understanding of the establishment and identification of phytogeographical regions in Gansu. The study primarily addresses the following key research questions: (1) To what extent are phylogenetic dissimilarity and taxonomic dissimilarity patterns in the Gansu flora interrelated? (2) What are the patterns of spatial turnover in species composition and phylogenetic composition between floras in Gansu? (3) Does the delimitation of floristic regions of Gansu incorporating phylogenetic information agree with previous biogeographic studies of the floristic region? (4) Where are the ranges and intersection boundaries for the major phytogeographic regions of China across this region?

## 2. Results

### 2.1. Beta Diversity

Comparing the sensitivity of our results using Mantel tests, we observed a strong correlation between β_jtu_ and β_sim_ (*r* = 0.99; *p* < 0.001). Similarly, there was a strong correlation between pβ_jtu_ and pβ_sim_ (*r* = 0.99; *p* < 0.001). Our focus was on the turnover component of the Sørensen index, specifically β_sim_ and pβ_sim_. We found a strong correlation between pβ_sim_ and β_sim_ (*r* = 0.96; *p* < 0.001). However, SES.pβ_sim_ exhibited a moderate correlation with pβ_sim_ (*r* = 0.25, *p* < 0.001) and no significant correlation with β_sim_ (*r* = 0.02; *p* < 0.31).

The close correlation between phylogenetic dissimilarity and taxonomic dissimilarity in Gansu’s flora was illustrated by the general spatial congruence in phylogenetic turnover and taxonomic turnover (Figure 1). The highest taxonomic and phylogenetic turnover occurred in the northwestern part of Gansu, and the lowest taxonomic and phylogenetic turnover occurred in the southeastern part of Gansu.

The null model test indicated that northwestern Gansu exhibited the highest regional phylogenetic turnover, while the southeast region showed the lowest phylogenetic turnover in Gansu (Figure 2). Among the 80 counties, the SES values were positive in 59 counties and negative in 21 counties, and SES values in 3 counties significantly differed from the null expectation (i.e., SES values > 1.96) (Figure 2).

### 2.2. Floristic Regions

Among the eight clustering algorithms tested, UPGMA consistently demonstrated the most effective alignment between the dendrogram and the original distance matrices for both β_sim_ and pβ_sim_ (Table 1). Thus, we selected the UPGMA method to define the phytogeographical regions. Despite a strong correlation between the β_sim_ and pβ_sim_ values (*r* = 0.96, *p* < 0.001), it is noteworthy that the distance matrix for pβ_sim_ has a closer match to the geographical clustering of recognized taxa types compared to the distance matrix for β_sim_. As a result, we decided to utilize pβ_sim_ for the definition of phytogeographical regions.

Pairwise phylogenetic beta diversity (pβ) metrics were employed to quantify the degree of change in phylogenetic composition across species assemblages within Gansu. Analyses of combined taxa pβ_sim_ values identified a total of nine floristic regions that are nested within two larger realms (Appendix A) and quantified the phylogenetic relatedness among all pairs of realms and regions (Figure 3). To maintain the stability of regional names, a nomenclature system was adopted for each region based on the naming convention established by Zhang et al. (2014) [50]. The regions were classified as follows: (1) region I—the northern foothills of the Qilian Mountains, including six county units; (2) region II—the hinterland of the Hexi Corridor, which comprises fourteen county units in the entire Hexi region except for the northern foothills of the Qilian Mountains; (3) region III—the Lanzhou–Baiyin wilderness region, which covers nine county units; (4) region IV—the Loess Plateau in the central region of Gansu, which includes eleven county units; (5) region V—the Loess Plateau in the east of Gansu, which encompasses thirteen county units; (6) region VI—the western Qinling Mountains, which comprises thirteen county units; (7) region VII—the transitional zone from Gannan Plateau to Longnan Mountainous Region, which includes six county units; (8) region VIII—the Gannan Plateau, which covers three county units; and (9) region IX—the Longnan Mountainous Region, which comprises five county units (Figure 3).

The application of pβ_sim_ dissimilarity matrixes resulted in the phytogeographical regions being partitioned into two prominent clusters in the dendrogram, with the northwest regions constituting a distinct cluster (Northwest) and the remaining southeast regions forming another separate cluster (Southeast) (Figure 3a,b). Given that these two major clusters were found to be located on either side of the Yellow River, they are commonly referred to as the Hexi floristic and the Hedong floristic, respectively.

The NMDS ordination method demonstrated a satisfactory projection of the dissimilarity matrixes of the Gansu floristic regions in two-dimensional space, as evidenced by the relatively low stress values (stress = 0.1064) (Figure 3c). The results of NMDS analysis are consistent with the results of cluster analysis, indicating that our division of the flora is feasible and reliable.

## 3. Discussion

### 3.1. Correlation between Phylogenetic and Taxonomic Patterns

We have presented an analysis of the spatial turnover of Gansu floristic assemblages in species composition and phylogenetic composition to examine the correlations between taxonomic dissimilarity and phylogenetic dissimilarity in this area.

The findings from this study revealed a strong correlation between taxonomic and phylogenetic turnover within the floristic assemblages of Gansu. Moreover, southeastern Gansu showed the lowest level of spatial turnover in both phylogenetic relationships and the taxonomic composition of floristic assemblages (Figure 1). The species turnover rate is low when the relief is complex, and the diversity of climates is high because the species are adapted to specific environmental conditions and are not able to survive in other conditions [55]. The complex relief and high diversity of climates create many different habitats that allow for a greater number of species to coexist without competing with each other [56]. The southeastern part of Gansu is located at the intersection of the western Qinling Mountains and the Qinghai–Tibet Plateau (QTP). It also serves as a transitional zone between the pan-Arctic plant region and the East Asian plant region, and it is connected to the Hengduan Mountains, which is a center of plant diversity in China [57]. The complex terrain and diverse humid climate in this region make it a transitional climate zone between subtropical and temperate zones, which is conducive to the formation of species diversity and provides a “refuge” for many species of ancient plant communities, where they are protected from the detrimental effects of severe weather [58]. Moreover, a large number of plant groups in this area have undergone geographical isolation due to the blocking effect of mountains, rivers, and gorges, which prevents the exchange of genetic information between individuals and thus increases the diversity of plant system development [59]. This may partly explain why floristic assemblages have on average a lower degree of spatial turnover in phylogenetic relatedness and taxonomic composition in southeast Gansu than in other regions in Gansu. This indicates that this area is a center of wild seed plant diversity in Gansu.

pβ_sim_ was strongly correlated with β_sim_, which implies that raw phylogenetic beta diversity is strongly influenced by the underlying patterns of species beta diversity. The results of the standardized effect size of pβ_sim_ (SES.pβ_sim_) show that phylogenetic beta diversity, except for some areas in the southeast, is higher than expected in most areas of central and western Gansu, meaning that the taxa turnover between floras in the vast majority of Gansu occurs between distantly related taxa (Figure 2). The pronounced heterogeneity of geographical environments is likely to give rise to significant differentiation and divergence in plant speciation and evolution across distinct floristic regions [60]. However, the potential correlation between plant species turnover, encompassing replacement and overlap, and the intricate geographical environments of Gansu within phylogenetic lineages necessitates further investigation.

Additionally, within the aforementioned regions, the Altun–Qilian Mountains region exhibited the most evolutionarily unique flora, featuring genera that displayed a significantly greater degree of phylogenetic turnover than anticipated (SES.pβ_sim_ > 1.96, Figure 3). The plants in the Altun–Qilian Mountain region originated during the Early Cretaceous period [61,62]. As the Tethys Sea receded and geological and climatic conditions rapidly deteriorated, the rate of plant differentiation increased, and plant species that were resilient to drought, salinity, and harsh environmental conditions were only able to survive [61]. The turnover of preserved plant species occurred at distantly related branches, leading to the formation of typical desert plant groups, such as Ephedraceae, Rhamnaceae, Chenopodiaceae, Zygophyllaceae, Plantaginaceae, etc. [61].

### 3.2. Phylogenetic Delimitation of Floristic Regions

Using all native seed plant genera that occur in Gansu, we identified two floristic regions (northwest cluster and southeast cluster) and nine subregions based on their phylogenetic affinities (Figure 3).

The prominent split of the primary two clusters indicates the presence of an east–west split within the floristic region of Gansu (Figure 3a). The floristic composition of the eastern region may have been shaped by the influences of both subtropical and temperate climates, while the western region may have been subjected to the impact of desert climate on vegetation [63,64]. This interpretation is supported by the phylogenetic relationships of the plant species, which also may reflect the environmental factors that have influenced their evolution and distribution. From a geographic and elevational perspective, the Wushaoling Mountains are a significant boundary for the differentiation of the Gansu flora between the eastern and western regions. The mountain range is situated at the intersection of the Loess Plateau, Qinghai–Tibet Plateau (QTP), and Inner Mongolia Plateau. It is a convergence point for three major climatic zones, namely, the high-altitude arid zone, the semi-arid zone in the temperate zone, and the arid zone in the temperate zone [65]. Moreover, it is the Wushaoling Mountains that serve as a boundary between the temperate monsoon climate and the temperate continental climate, and it is also a transition zone from the semi-arid zone to the arid zone [66].

Our research not only captured the two broader phytogeographic differentiations among plant assemblages within Gansu Province but also embraced the subtler ecological nuances inherent in its botanical composition. The incorporation of nine subordinate subregions within the overarching biogeographic classification serves to underscore the intrinsic ecological intricacy permeating Gansu Province. We delineated Gansu Province into multiple levels of biogeographic divisions, encompassing both broader and more localized scales, to provide a holistic understanding that reflects the true nature of the province’s biotic diversity and its historical biogeographic significance.

From the nine phytogeographical regionalization schemes obtained (Figure 3), we found these floristic regions to be similar to the previously published floristic division of Gansu based on qualitative evidence provided by experts and the modeled distribution of vascular plant species [50]. However, the range and boundary of the flora that we found are significantly different from previous studies. First, the northwestern Gansu region is separated into regions I and II, instead of being a single floristic region as in previous classifications. The plant communities in the northwest region, although belonging to the same larger plant group, have evolved and adapted differently over a long period, resulting in the differentiation of two distinct subgroups in the systematics between the northern foothills of the Qilian Mountains and the interior of the Hexi Corridor. Warm and humid Pacific air currents are blocked by the Qinghai–Tibet Plateau and the Qilian Mountain Range, gradually resulting in an arid climate in the Hexi Corridor [63,67] and eventually forming an extremely arid desert and plant geographical system [68]. However, the significant variation in altitude and the complex environment formed by water from melting snowmelt in the Qilian Mountains have led to the differentiation or specialization of plant species [50], resulting in a marked increase in species richness and endemism in the Qilian Mountains region compared to the plains and desert areas of the Hexi Corridor. Secondly, region VI is a single unit in our floristic region while Zhang et al. (2014) [50] divided it into two distinct regions. This region is situated in the western Qinling Mountains and occupies a transitional zone between the subtropical and warm temperate plant floras, and the result is that this region is characterized by a wide range and compositions that include multiple elements [69].

Notably, the clustering tree, based on phylogenetic relationships, revealed an extended southern distribution range of desert plants in region III. This region is located on the northern edge of the Loess Plateau, and it has long been affected by the arid desert climate of the Alashan region [70]. This result aligns with the observations made by Wu et al. [47], which indicate that Lanzhou serves as the southeastern boundary for the distribution of desert plants in China. The area of region IV is smaller than the geographic range of the central Loess Plateau region in Gansu, indicating a significant discrepancy. This disparity may be attributed to the fact that the biotic diversity pattern of the Loess Plateau region is predominantly influenced by the north–south dispersal of species, and differentiation does not play a dominant role in shaping the pattern [71]. It also should be noted that plant species in zone VI have expanded northward to the Liupan Mountain region (Huating and Pingliang) (Figure 3b). This may be related to the geographic partitioning effects of the famous mountain ranges, namely the Qinling and Liupan Mountains [59].

Previous studies revealed that the biodiversity patterns in Gansu were shaped by historical events, such as the uplift of the QTP and the Hengduan mountain from the end of the Tertiary period to the beginning of the Quaternary period [44,47,72]. The dendrogram generated by our pβ_sim_-matrix-based clustering analysis depicts the phylogenetic relationships among the floras of different regions and underscores the importance of shared evolutionary histories, which encompass origin, diversification, and dispersal events, in shaping the spatial distribution patterns of plant species in Gansu [5,33]. The observed dissimilarities provide compelling evidence that the phylogenetic dissimilarity approach is an effective tool for capturing the evolutionary history of the Gansu flora, thereby furnishing greater insights than the taxonomic dissimilarity approach into the interrelationships among different phytogeographical regions [5].

Our findings showed that the phylogenetic dissimilarity approach, which relies on a comprehensive and robust regional phylogeny at the genus level, is a superior method for uncovering concealed phylogenetic relationships between biogeographic regions compared to the taxonomic dissimilarity approaches. This regionalization based on quantitative clustering algorithms could reveal the optimal number of regions and yield clear boundaries, which is more rational than relying on experts to qualitatively delineate the number of recognized regions and their boundaries [5,12,22].

### 3.3. Ranges and Intersection Boundaries of China’s Four Major Phytogeographic Regions

Ye et al. (2019), in a study on the β-diversity of angiosperm phylogeny in China, grouped the QTP region and the Tethyan region together [5], which is consistent with the results of our study (northwest cluster) (Figure 3a). Based on the research conducted by Ye et al. (2019) regarding the phylogenetics of angiosperms in China, it was observed that the phytogeographic composition of Gansu Province is precisely situated at the confluence of four major phytogeographic realms within China, namely the Holarctic, the East Asian, the Tethyan, and the QTP regions (Figure 4). Specifically, (1) regions I, VII, and VIII belong to the QTP floristic region, with region I located at the northern edge of QTP, and regions VII and VIII are situated on the eastern edge of QTP; (2) region II belongs to the Tethyan floristic region, with rich components of the Tethyan flora that evolved differently from those of the Holarctic and East Asian regions but have a closer relationship with the QTP flora [5]; (3) regions III–V belong to the Holarctic flora region, with rich components of the Holarctic Tertiary flora and a close relationship to the East Asian floristic region [5]; (4) region IX belongs to the East Asian floristic region, preserving the once widely distributed tropical flora of the Northern Hemisphere that became extinct in other areas due to the influence of the Quaternary ice age [73,74,75].

The intricate nexus of phytogeographic regions in Gansu Province exhibits profound significance [15]. The amalgamation of these regions engenders a heterogeneity of biotic elements, resulting in a region that is characterized by elevated richness and the intermingling of plant taxa [76]. This conspicuous convergence brings forth the juxtaposition of diverse flora, each with its own historical trajectory, ecological adaptation, and evolutionary trajectory, thereby catalyzing a mosaic of ecological niches and selective pressures that underpin the observed botanical diversity [77]. In summary, the positioning of Gansu Province as an intricate phytogeographic nexus bears profound implications for botanical diversity. Its status as an intersection of realms amplifies the significance of this region in the discourse of plant evolution, biogeography, and ecological adaptation. The intricate interplay of these realms within Gansu Province warrants sustained scholarly attention, promising a deeper comprehension of the underlying drivers and consequences of this unique botanical confluence.

From an evolutionary perspective, this study’s findings open new avenues for exploring floristic assemblages in Gansu province: (1) Investigating historical biogeographic processes that shaped plant diversity and distribution patterns by tracing lineage evolution and geographic spread can reveal key factors driving current trends. (2) Future studies can uncover how phylogenetic relationships influence community assembly, including the roles of ecological filters, competition, and coexistence mechanisms. This sheds light on the evolutionary dynamics of local plant communities. (3) Research into how specific environmental factors, like climate and topography, influence lineage diversification in Gansu offers insights into the interplay between ecology and evolution. (4) Comparing Gansu with diverse regions can elucidate general patterns, unique features, and the relative impact of historical processes and contemporary factors on plant diversity. (5) Utilizing phylogenetic regions for conservation planning enhances the protection of areas with high phylogenetic diversity, endemism, and evolutionary distinctiveness, preserving both species and evolutionary history.

## 4. Materials and Methods

### 4.1. Study Area

Gansu Province is a long and narrow region in northwest China, occupies an area of 453,700 km^2^, and is located between 32°11′– 42°57′ N and 92°13′–108°46′ E, with elevations ranging from 526 to 5773 m asl (Figure 5) [78]. It has a complex and diverse physical geography and climate, as it lies at the intersection of three major plateaus: the Loess Plateau, the Inner Mongolia Plateau, and the Qinghai–Tibet Plateau [49]. It also spans four major climatic types based on the Rivas-Martínez World Bioclimatic classification: warm temperate continental and cold temperate continental [79]. It is also an important water conservation and supply area in the upper reaches of the Yangtze and Yellow Rivers [77]. The geological, topographical, and climatic diversity of the region is believed to have played a crucial role in the development and establishment of the flora in this area [77].

### 4.2. Species Distribution Data

The species distribution data were diligently compiled from national and regional flora reports, checklists, scholarly literature, and herbarium records (a full reference list is provided in Appendix A). The data were purged of duplicates and records that lacked reliable georeferencing to the level of the county. Excluding non-native species, the final matrix consisted of 5244 records belonging to 1131 genera and 170 families. The names of all species were standardized using the R [80] software package ‘plantlist’ [81], with the order of families based on the nomenclature of Angiosperm Phylogeny Group IV [82].

To analyze the spatial changes in species composition and phylogenetic relatedness, the study area was divided into 80 county-level geographical units (Appendix A). Due to limitations in available data, taxonomic uncertainties, and hybridization events, species-level identification poses challenges. Hence, we decided to conduct the analysis at the genus level, in accordance with Li et al. (2015) [4] and Ye et al. (2019) [5]. The presence or absence of each genus of seed plants in each county was ascertained by extracting the distribution information of each genus and constructing a presence–absence matrix (Appendix A).

### 4.3. Phylogeny Construction

A phylogenetic tree at the genus level was generated by incorporating the genera within our study area into an existing backbone phylogenetic hypothesis using the ‘V.PhyloMaker’ package [83] in R software. For genera that were not present in the mega-tree, a placement was assigned based on a closely related genus using scenario 3’s options for phylogeny generation [83]. Considering the limited availability of comprehensive time-calibrated phylogenies for families and genera, we followed a similar methodology as previous studies and treated undetermined genera as polytomies located within their respective families [4,84]. The constructed phylogenetic tree was visualized using the Interactive Tree of Life (iTOL) online software application (https://itol.embl.de (accessed on 20 November 2022 )) (Appendix A).

### 4.4. Phylogeny Diversity Analyses

To quantitatively evaluate the spatial turnover of floristic composition in Gansu, we utilized the Simpson dissimilarity index (β_sim_) [85]. The β_sim_ index allows us to assess the taxonomic turnover that occurred between each pair of regions. The calculation of β_sim_ (1) is performed as follows:(1)$$\beta_{sim}=\frac{\min\left(b,c\right)}{a+\min\left(b,c\right)}$$
where ‘a’ represents the count of shared genera between two regions, while ‘b’ and ‘c’ correspond to the counts of genera that are unique to each respective region. The values of β_sim_ range from 0 to 1, with 0 indicating an identical generic composition between the regions and 1 denoting the absence of shared taxa between the regions [65,85]. One notable advantage of the β_sim_ metric is its ability to mitigate the impact of species richness heterogeneity across different regions, thus ensuring a robust assessment of compositional turnover [85].

To quantify the phylogenetic turnover in our study, we utilized phylogenetic beta diversity (pβ_sim_), an adapted version of the β_sim_ index. Instead of considering the proportion of shared genera, pβ_sim_ assesses the proportion of shared phylogenetic branch lengths in the dated phylogenetic tree between regions [44]. The values of pβ_sim_ range from 0 to 1, where 0 indicates that the genera have identical branch lengths and share the same evolutionary history, while 1 indicates that the genera have distinct branch lengths and do not share similar evolutionary trajectories. Pairwise distance matrices were calculated for both pβ_sim_ and β_sim_ across all regions. Additionally, we compared the obtained values of β_sim_ and pβ_sim_ with those derived from the Jaccard Index (β_jaccard_) to assess the sensitivity of our results (β_jtu_ and pβ_jtu_) [86].

To investigate the potential association between pβ_sim_ and β_sim_, we employed a null model approach to assess whether county-level assemblages exhibited greater or lesser phylogenetic similarity compared to what would be expected based solely on the number of taxa. This null model analysis involved generating a null distribution by randomly permuting the tips of the phylogenetic tree for a total of 999 times. For each permutation, pairwise distance matrices were computed for both pβ_sim_ and β_sim_, capturing the phylogenetic turnover between all regions while preserving the inherent taxonomic turnover differences among them. Subsequently, pβ_sim_ values were calculated for each pair of compared regions by generating pairwise distance matrices for pβ_sim_ [5]. Standardized effect sizes of pβ_sim_ (SES.pβ_sim_) (2) were obtained by comparing the observed pβ_sim_ value to the mean of the null distribution and dividing it by the standard deviation of the null distribution. Positive values of SES.pβ_sim_ indicate a higher level of phylogenetic turnover than expected under random sampling, while negative values indicate the opposite scenario.
(2)$$SES.p\beta_{sim}=\frac{obs.p\beta_{sim}-mean(p\beta_{sim}.null)}{sd.(p\beta_{sim}.null)}$$

Statistical analyses were performed using the ‘betapart’ package in R [86] to examine the patterns of regional turnover in phylogenetic composition. For each county, we calculated the mean values of β_sim_, pβ_sim_, and SES.β_sim_ compared to other counties. These values were then mapped to visually represent the spatial patterns of phylogenetic composition turnover [5,12]. Furthermore, we conducted Mantel tests to assess correlations between species beta diversity, phylogenetic beta diversity, and SES.β_sim_. The Mantel tests allowed us to investigate the associations between distance matrices [87]. To determine the significance of the correlation, permutation tests were employed by randomizing the distance matrix 999 times. All statistical analyses were performed in R using the ‘vegan’ package [88] and the ‘picante’ package [89].

### 4.5. Cluster Analysis

In order to define the floristic regions of Gansu by incorporating distribution data and phylogenetic information, we applied hierarchical clustering to β_sim_ and pβ_sim_ pairwise distances. Given that the choice of clustering algorithms and linkage functions can greatly impact the results, we initially assessed the performance of eight linkage functions in agglomerative hierarchical clustering. The linkage functions employed in this analysis included single linkage (SL), complete linkage (CL), unweighted pair-group method using arithmetic averages (UPGMA), unweighted pair-group method using centroids (UPGMC), weighted pair-group method using arithmetic averages (WPGMA), weighted pair-group method using centroids (WPGMC), and two variations of Ward’s minimum variance (ward.D and ward.D2). All of these linkage functions were implemented using the ‘cluster’ package in R.

We evaluated the soundness of the clustering outcomes by employing Sokal and Rohlf’s (1962) [90] cophenetic correlation coefficient and Gower’s (1983) [91] distance. The cophenetic correlation coefficient gauges the correlation between the terminal branches of the dendrogram and the original distance matrix with a range of 0 to 1, wherein higher values indicate a stronger correlation. Gower’s distance computes the sum of squared discrepancies between the original distances and the cophenetic distances. A clustering method that minimizes Gower’s distance is deemed suitable for the distance matrix [92]. To ascertain a reasonable number of clusters for the phytogeographical regions, we employed the “Silhouette method”. The Silhouette method is an interpretive and validation approach that provides a concise graphical representation of the classification effectiveness for each object, assessing the consistency within clusters of data [72,93]. The ‘Silhouette’ method was executed using the “silhouette” function within the “cluster” package in R.

We employed three criteria to determine the number of clusters and identify the floristic regions based on the clustering results: (1) a preference for the contiguous aggregation of sites to represent a floristic region, (2) the requirement for each cluster representing a floristic region to form a monophyletic clade in the dendrogram, and (3) consistency between the identified floristic region and the geography of Gansu.

### 4.6. Ordination Analysis

To provide an alternative non-hierarchical depiction of cluster relationships, a two-dimensional ordination was conducted to visualize the floristic regions of Gansu. This was accomplished by utilizing a neighbor-joining algorithm in combination with the non-metric multidimensional scaling (NMDS) method, which is widely regarded as a reliable unconstrained ordination approach for representing the overall turnover values within a matrix in a low-dimensional space [94]. The NMDS ordination was performed using the pβ_sim_ pairwise distance for cluster-defined floristic regions, employing the “vegan” package in R and utilizing one hundred random starts to ensure a stable solution and to mitigate the influence of local minima. The stress value is commonly employed to evaluate the correspondence between the NMDS and the original dissimilarity matrix. Stress values range from 0 to 1, with lower values indicating better NMDS results.

## 5. Conclusions

In this study, a novel approach utilizing phylogenetic relationships was employed to delineate the floristic regions within Gansu. This approach provided a more comprehensive understanding of the biogeographic patterns and evolutionary history of plant lineages within the region by incorporating information on the shared ancestry and diversification history of taxa. The resulting phylogenetic delineation of floristic regions resulted in a framework for investigating the historical biogeographic processes and environmental factors that presented the patterns of plant diversity and distribution in Gansu. These findings suggest that historical biogeographic processes have played a crucial role in shaping the current patterns of plant diversity and distribution in these areas. By accounting for the phylogenetic relationships among plant lineages, we can gain a deeper understanding of the underlying ecological and evolutionary mechanisms that have led to the observed patterns of phylogenetic turnover and beta diversity across the study region.

## Figures and Tables

**Figure 1 plants-12-03060-f001:**
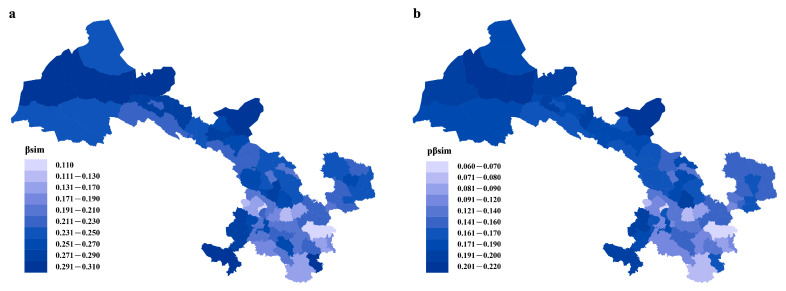
Patterns of spatial turnover of Gansu flora. Numbers indicate the average values of β_sim_ (**a**) and pβ_sim_ (**b**) for the focal region and all other regions. The maps were generated using ArcGIS 10.8.

**Figure 2 plants-12-03060-f002:**
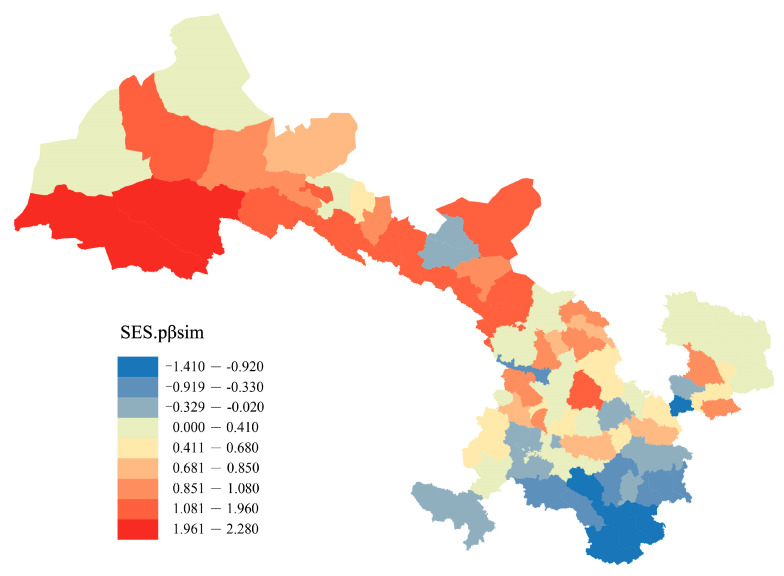
Patterns of spatial turnover of Gansu flora. The numbers are the mean standardized effect sizes of phylogenetic beta diversity (SES.pβ_sim_) for each county compared to all other counties. Positive values indicate regions with higher phylogenetic turnover than expected, whereas negative values indicate the opposite.

**Figure 3 plants-12-03060-f003:**
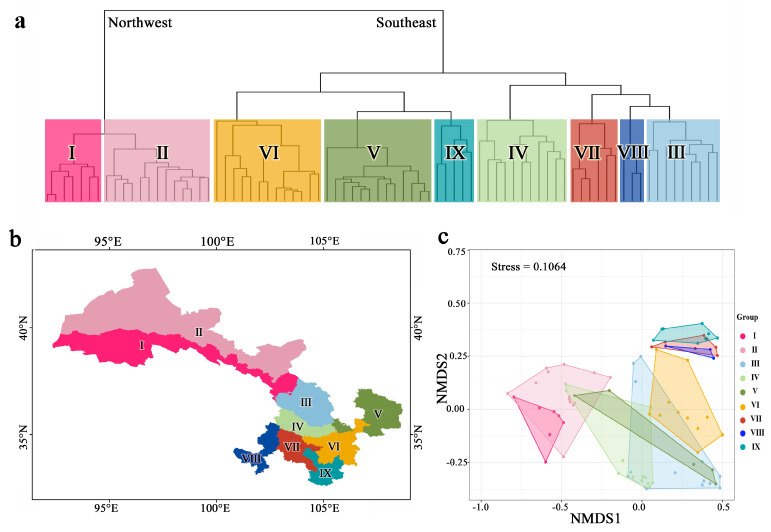
Dendrogram (**a**) and map (**b**) resulting from UPGMA hierarchical clustering and the NMDS ordination of grid cell assemblages based on phylogenetic beta diversity distance matrixes at the genus level (**c**). The dendrogram highlights the nine phytogeographical regions, which are also represented by corresponding colors in the NMDS ordination. Additionally, the map, generated using ArcGIS 10.8, visually displays the spatial distribution of these regions. I—the northern foothills of the Qilian Mountains; II—the hinterland of the Hexi Corridor; III—the Lanzhou–Baiyin wilderness region; IV—the Loess Plateau in the central region; V—the Loess Plateau in the east region; VI—the western Qinling Mountains; VII—the transitional zone from Gannan Plateau to Longnan Mountainous Region. VIII—the Gannan Plateau; and IX—the Longnan Mountainous Region.

**Figure 4 plants-12-03060-f004:**
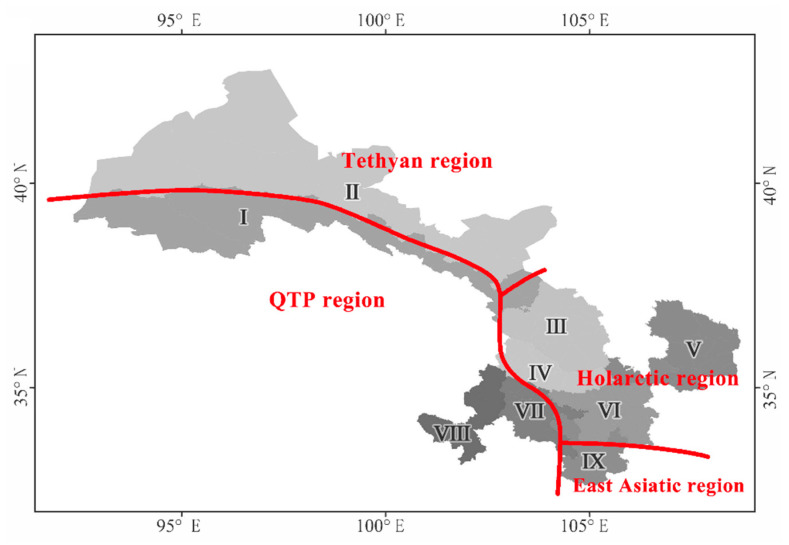
The position of Gansu flora in the flora of China based on pβ_sim_ dissimilarity. I—the northern foothills of the Qilian Mountains; II—the hinterland of the Hexi Corridor; III—the Lanzhou–Baiyin wilderness region; IV—the Loess Plateau in the central region; V—the Loess Plateau in the east region; VI—the western Qinling Mountains; VII—the transitional zone from Gannan Plateau to Longnan Mountainous Region. VIII—the Gannan Plateau; and IX—the Longnan Mountainous Region.

**Figure 5 plants-12-03060-f005:**
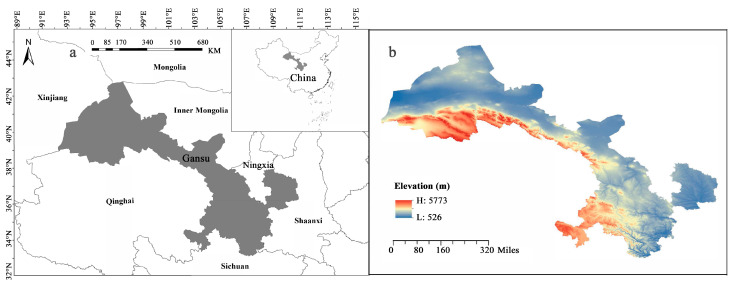
Map showing the location of Gansu Province in Northwest China (**a**) and elevation information (**b**). Gansu Province is in northwest China and adjacent to Qinghai, Sichuan, Shaanxi, Inner Mongolia, Ningxia, and Xinjiang. There are large differences in altitude.

**Table 1 plants-12-03060-t001:** Cophenetic correlation coefficient (CCC) and Gower’s distance (GD) for eight different clustering methods based on β_sim_ and pβ_sim_ pairwise distance matrixes.

Clustering Algorithms	CCC (pβ_sim_)	GD (pβ_sim_)	CCC (β_sim_)	GD (β_sim_)
ward.D2	0.840	13,784.600	0.860	13,779.820
ward.D	0.814	213,347.500	0.830	212,874.300
single	0.856	298.473	0.838	335.790
Complete	0.859	266.409	0.869	230.959
UPGMA	0.872	38.090	0.876	37.130
WPGMA	0.867	44.907	0.873	38.218
WPGMC	0.813	243.142	0.830	409.498
UPGMC	0.813	213,347.500	0.830	212,874.300

## Data Availability

The original contributions presented in the study are included in the Appendix A.

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
