# Peer review of "Phylogenetic Partitioning of Gansu Flora: Unveiling the Core Transitional Zone of Chinese Flora"

_plants, 2023, doi:10.3390/plants12173060_

Round 1

Reviewer 1 Report

Comments to the manuscript entitled “Phylogenetic partitioning of Gansu Flora: Unveiling the Core Transitional Zone of Chinese Flora”

1.     This manuscript proposes to dissect the biogeography of the Core transitional zone of Chinese flora using a phylogenetic approach and compare it to a divergence approach (beta-diversity). It is well-written, and its objectives are relevant for biogeographic analyses, but I think that there are several areas of the manuscript that need justification and clarification. Usually, this approach uses a geographic unit related to administrative definition, in this case counties, it is based in ecological division of the territory with the presence or absence of a particular genus. In consequence the paired turnover for counties expresses the proportion of genera change in each case.  I have both general and specific comments.

General comments:

2.     There is a lack of comparisons with more traditional approaches like the use of the principle of taxonomic endemism to define biogeographic areas. Some of the approaches could be used as “proof of concept” for the phylogenetic data used. There are other publications that have partially done this and also they could be cited.

3.     Usually, these analyses uses species as taxonomic units. In this manuscript, genera were used. Why was this used is not justified. Either species or families could be used instead. This needs further clarification either using Gensu flora or comparisons made in published literature (e.g. Ye et al. 2019. Molecular Phylogenetics and Evolution or Swenson, 2011, PLOSOne)

Specific comments

4.     Lines 38-95. I understand that since Wallace and Darwin there is a relationship between biogeographic patterns and phylogenetic ones. The problem is that there are different possible alternatives. For example, phylogeny could be dispersed in different biogeographic areas (Swenson, 2011). This is probably because processes in both phylogenies and biogeographies are overlapping but not always. When you use genera as taxonomic units these anomalies are blurred.

5.     Lines 98-100, explain why the four plant subregions already defined and using the seven geomorphic types are not sufficient to define the biogeography of Gansu. See also comments on the discussion section

6.     Lines 113-116, and lines 121-124,  it would be important to be more specific about the uses of the phytogeographic regionalization and how does it reinforces regionalizations using other groups of species

7.     Lines 124-126, theoretically how would both dissimilarities be associated?

8.     Lines 129-130, I am not sure that this question was answered

9.     Fig. 3, it is not clear how the expected estimates were obtained, clarify

10.  Lines 202-205, Why would not define just two biogeographic regions (I and II) and III to IX, this has to be included as an alternative within the discussion section

11.  Lines 211-212, why would there by a high or low correlation between taxonomy and phylogeny?

12.  Lines 214-216, this implication appears already known and published and it can be deleted

13.  Lines 257-355, why is it necessary to define phylogenetically floristic regions if the taxonomically defined are so similar, please justify

14.  Lines 350-355, this paragraph is too general, it could be deleted

15.  Lines 490-491, note that while the approach is novel, the results change very little the biogeography of the flora that has been already defined, include this in the conclusions

16.  Lines 499-501, this assertion is obvious and it could be deleted

17.  Lines 505-533, this paragraph belongs to the discussion section and also the ideas should be included and properly referenced in the introduction

Author Response

Dear Editor and dear reviewers,

Re: Manuscript ID: “plants-2503187” and Title: “Phylogenetic Partitioning of Gansu Flora: Unveiling the Core Transitional Zone of Chinese Flora”

Thank you for your letter and the reviewers’ comments concerning our manuscript entitled “Phylogenetic Partitioning of Gansu Flora: Unveiling the Core Transitional Zone of Chinese Flora” (ID: plants-2503187). Those comments are valuable and very helpful. We have read through comments carefully and have made corrections. Based on the instructions provided in your letter, we uploaded the file of the revised manuscript. Revisions in the text are shown using red highlight for additions, and strikethrough font for deletions. The responses to the reviewer's comments are marked in blue and presented following.

We would love to thank you for allowing us to resubmit a revised copy of the manuscript and we highly appreciate your time and consideration.

Sincerely.

Zizhen Li.

Reviewer #1:

Q1: There is a lack of comparisons with more traditional approaches like the use of the principle of taxonomic endemism to define biogeographic areas. Some of the approaches could be used as “proof of concept” for the phylogenetic data used. There are other publications that have partially done this and also they could be cited.

Response: Thank you for your suggestion. According to reviewer's comment, we have refined and revised the introduction in the revised manuscript (Lines 41-47 and lines 93-103). As follows:

Qualitative methods based on taxonomic dissimilarities, encompassing species, generic, and familial levels, have been applied to the process of biogeographical regionalization. Several biomes and biogeographical regionalization schemes have been proposed, such as the world’s flora, East Asian Plants, and the flora of China. These schemes offer a spatially explicit framework to investigate biodiversity patterns on a large scale.

The phylogenetic approach and traditional methods in biological research exhibit distinct methodologies and focus areas. Traditional methods rely on observable traits, historical taxonomic classifications, and characteristics, which as morphology, ecology, and behavior, to infer relationships, ecological interactions, and species identities. The phylogenetic approach centers on analyzing genetic diversity and evolutionary history among organisms using molecular data, revealing shared ancestry and divergence times. Integrating these approaches can offer a more comprehensive understanding of biodiversity, blending genetic insights with traditional ecological and taxonomic perspectives. Quantitative methods of phylogenetic information, rather than qualitative methods that rely on experts’ experience and knowledge, are more objective and repeatable.

Q2: Usually, these analyses use species as taxonomic units. In this manuscript, genera were used. Why was this used is not justified. Either species or families could be used instead. This needs further clarification either using Gensu flora or comparisons made in published literature (e.g. Ye et al. 2019. Molecular Phylogenetics and Evolution or Swenson, 2011, PLOSOne)

Response: Thank you for your comments, the discussion regarding this question is presented following:

Phylogenetic divisions based on family-level phylogeny may be helpful for rough or preliminary phylogenetic divisions, but more detailed analysis requires phylogenetic divisions at the genus or species level, especially for plants with complex evolutionary history. The choice of taxonomic units in ecological and evolutionary analyses, such as those conducted in the manuscript, often depends on the specific research question, available data, and scientific rationale. In this case, the manuscript departs from the convention of using species as taxonomic units and instead employs genera for its analyses. The use of genera as taxonomic units could be justified if it aligns with the goals of the study. Genera might offer a balance between species-level resolution and higher-level taxonomic grouping, providing a compromise that captures both the diversity within and between species. This can be particularly relevant when species-level identification is challenging due to limited data availability, taxonomic uncertainties, or hybridization events. By analyzing at the genus level, the study aim to uncover broader patterns in ecological and evolutionary processes while avoiding potential noise or ambiguity associated with species-level identification. According to the reviewer’s comment, we have added a more detailed interpretation regarding in lines 189-196. As follows:

To analyze the spatial changes in species composition and phylogenetic relatedness, the study area was divided into 80 county-level geographical units (Supplementary Figures S1, Supplementary Table S1). Due to limitations in available data, taxonomic uncertainties, and hybridization events, species-level identification poses challenges. Hence, we decided to conduct the analysis at the genus level, in accordance with Li et al (2015) and Ye et al (2019). The presence or absence of each genus of seed plants in each county was ascertained by extracting distribution information of each genus and constructing a presence-absence matrix (Supplementary data matrix).

Q3: Lines 38-95. I understand that since Wallace and Darwin there is a relationship between biogeographic patterns and phylogenetic ones. The problem is that there are different possible alternatives. For example, phylogeny could be dispersed in different biogeographic areas (Swenson, 2011). This is probably because processes in both phylogenies and biogeographies are overlapping but not always. When you use genera as taxonomic units these anomalies are blurred.

Response: Thank you for underlining this deficiency. Due to various factors such as aged specimens, limited specimen information, insufficient field sampling, and inadequate establishment of the Gansu Plant Data Sharing Platform, the availability of data at a finer resolution, specifically at the species level, is constrained. In order to mitigate these limitations, we opted for an analysis at the genus level. From the outcomes obtained in our study, it is evident that the data resolution at the genus level adequately supports the principal objectives and contents of our research. Furthermore, the results align with established patterns and theoretical expectations, thus affirming the feasibility of conducting our research at the genus level. The forward-looking issues you raised are also under consideration within the scope of our ongoing efforts. In order to make this difference more clear, we intend to undertake a more in-depth research by combining species-level data for Gansu endemic plants with environmental factors such as precipitation, mean temperature, and terrain variability.

Q4:  Lines 98-100, explain why the four plant subregions already defined and using the seven geomorphic types are not sufficient to define the biogeography of Gansu. See also comments on the discussion section.

Response: Thank you for your comments, the discussion regarding this question is presented following:

The traditional approach to defining plant biogeographic regions based solely on species diversity methods may fall short in capturing the full extent of evolutionary relationships among species. This is why scholars often seek to complement the existing definitions with phylogenetic diversity approaches. Here's the rationale behind this academic inclination:

  1. Traditional species diversity methods primarily rely on observable traits, ecological characteristics, and historical taxonomic classifications. However, these traits might not accurately represent the underlying evolutionary relationships among species. Phylogenetic diversity methods, by analyzing genetic data, unveil hidden evolutionary connections that cannot be discerned solely through observable traits.
  2. Traditional methods might encounter challenges when dealing with convergent evolution, where unrelated species develop similar traits due to similar environmental pressures. Such instances can lead to misinterpretations of evolutionary relationships and result in erroneous biogeographic classifications. Incorporating phylogenetic information can help differentiate true relationships from convergent traits.
  3. Phylogenetic diversity approaches provide a finer-scale resolution of relationships between closely related species, enabling researchers to distinguish among cryptic species and populations. This level of detail is often beyond the capabilities of traditional methods.
  4. Understanding the evolutionary history and relationships of species is crucial for designing effective conservation strategies. Phylogenetic approaches can highlight areas of high evolutionary diversity that might not be apparent using traditional species-based methods alone.
  5. Phylogenetic information provides insights into the evolutionary potential of species and their responses to changing environments. This knowledge is valuable for predicting how plant communities may adapt and change in the face of ongoing environmental shifts.
  6. In our manuscript, we used the GBOTB tree in 'V.PhyloMaker' package to generate our phylogenetic tree. The GBOTB tree is Smith and Brown (2018) published a phylogeny that was generated based on molecular data from GenBank and incorporated data from the Open Tree of Life project. Its advantages stem from its incorporation of molecular data and a phylogenetic framework, resulting in a more accurate and evolutionary-based classification of vascular plants.

Given these reasons, scholars engage in incorporating phylogenetic approaches to redefine plant biogeographic regions based on the existing definitions. By doing so, We aim to enrich the understanding of biogeography by considering both the historical relationships among species and their contemporary ecological attributes. The integration of phylogenetic methods adds depth, precision, and a dynamic evolutionary perspective to the traditional understanding of plant distributions, ultimately contributing to a more comprehensive and accurate delineation of biogeographic regions.

Q5: Lines 113-116, and lines 121-124, it would be important to be more specific about the uses of the phytogeographic regionalization and how does it reinforces regionalizations using other groups of species.

Response: Thank you for your comments, the discussion regarding this question is presented following:

  1. Phytogeographic regionalization provides essential insights into the distribution patterns and evolutionary history of plant species. This knowledge is crucial for designing effective conservation strategies by identifying areas of high species richness, endemism, or unique evolutionary lineages. By integrating data from different species groups, such as animals, fungi, and microbes, researchers can establish comprehensive conservation plans that consider the entire ecosystem.
  2. Understanding phytogeographic regions contributes to our knowledge of biogeographic evolution and historical changes in species distributions. By incorporating information from multiple species groups, researchers can trace the co-evolutionary relationships between plants, animals, and other organisms, revealing how these interactions have shaped the biogeography of the region over time.
  3. Phytogeographic regionalization aids in unraveling the intricate relationships between different species within an ecosystem. By studying the distribution of various species groups, researchers can gain insights into ecosystem functioning, species interactions, and the roles different organisms play in maintaining ecological balance.
  4. Integrating data from various species groups enhances taxonomic and phylogenetic research efforts. Comparative analyses of evolutionary histories across different taxa can lead to a more comprehensive understanding of the region's evolutionary processes. This approach can help identify key evolutionary events that have influenced the formation of distinct phytogeographic regions.
  5. Phytogeographic regionalization, when combined with data from other species groups, can facilitate predictive biogeography. Researchers can use this information to model potential future shifts in species distributions under changing environmental conditions. These predictive models aid in proactive conservation planning and management.

In summary, the phytogeographic regionalization of an area provides valuable insights into evolutionary, ecological, and conservation aspects of biodiversity. Integrating data from various species groups reinforces regionalizations by offering a more holistic understanding of the region's history, interactions, and potential future changes. This multidisciplinary approach enriches our comprehension of the complex dynamics that have shaped and continue to influence the biogeography of the studied area.

According to reviewer's comment, we have refined and revised the introduction in the revised manuscript (Lines 141-151). As follows:

Phytogeographic regionalization holds crucial significance in comprehending the evolutionary dimension of floristic diversity, particularly in the specified area. By investigating the distribution patterns and evolutionary histories of plant species, it offers insights into conservation strategies, biogeographic evolution, ecosystem functioning, and taxonomic research. To reinforce regionalizations, integrating data from various species groups, such as plants, animals, and microbes, helps unveil co-evolutionary relationships, predict ecosystem shifts, and enhance our understanding of the region's history and potential future changes. This multidisciplinary approach not only aids in delineating phylogenetically informed floristic regions but also deepens our grasp of phytogeographical establishment and identification, offering a comprehensive perspective on the complex dynamics underlying the biogeography of the region.

Q6: Lines 124-126, theoretically how would both dissimilarities be associated?

Response: Thank you for your comments, the discussion regarding this question is presented following:

Phylogenetic dissimilarity refers to the divergence in evolutionary relationships among species within different geographical regions or ecological communities. Taxonomic dissimilarity, on the other hand, reflects the degree of difference in species composition based on their classification into taxonomic groups. The extent of association between these two dissimilarities is influenced by several factors:

  1. Species within the same taxonomic group share a common evolutionary history to varying degrees. Taxonomic dissimilarity patterns might be driven by phylogenetic relationships; closely related species are more likely to be categorized within the same taxonomic group. However, exceptions arise due to factors like convergent evolution, hybridization, and taxonomic revisions.
  2. Ecological processes, such as adaptation to local environments, could result in similar species being present across different regions, leading to lower taxonomic dissimilarity despite considerable phylogenetic divergence. Conversely, closely related species might occupy distinct niches, contributing to high taxonomic dissimilarity and moderate phylogenetic dissimilarity.

3.Historical events like dispersal, vicariance, and extinction shape species distributions. Phylogenetic dissimilarity could reflect historical biogeographic events more directly, whereas taxonomic dissimilarity might be influenced by factors such as dispersal abilities and colonization history.

  1. Taxonomic classifications might not always accurately represent evolutionary relationships. Taxonomic revisions, lumping, or splitting of taxa can influence taxonomic dissimilarity patterns without necessarily impacting phylogenetic dissimilarity in the same way.
  2. Sampling intensity and spatial scale can affect observed patterns. Uneven species representation or scale differences might lead to inconsistencies between phylogenetic and taxonomic dissimilarities.

Theoretical associations between these dissimilarities can vary:

  1. In cases where evolutionary history closely aligns with taxonomic classification, phylogenetic and taxonomic dissimilarity patterns might be congruent. Regions with high phylogenetic dissimilarity could exhibit high taxonomic dissimilarity due to the accurate representation of evolutionary divergence in taxonomy.
  2. Phylogenetic and taxonomic dissimilarities might diverge due to ecological factors, historical biogeography, or taxonomic uncertainties. High phylogenetic dissimilarity could coincide with low taxonomic dissimilarity if niche-driven ecological processes lead to non-random species composition.
  3. Various scenarios can exist where partial congruence or discordance exists between the two dissimilarities, illustrating the complex interplay between evolutionary and taxonomic factors.

Ultimately, understanding the theoretical associations between phylogenetic and taxonomic dissimilarity patterns requires a holistic consideration of evolutionary, ecological, and historical dynamics. Empirical studies, incorporating robust phylogenetic reconstructions and comprehensive taxonomic assessments, are crucial to untangle the intricate relationship between these dissimilarities within the Gansu flora and other biological systems.

Q7: Lines 129-130, I am not sure that this question was answered

Response: We sincerely thank the reviewer for careful reading. we have improved and revised the issue in the revised manuscript (Lines 505-534). As follow:

Ye et al. (2019) in a study of the β-diversity of angiosperm phylogeny in China, the Holarctic and East Asian regions were clustered together in a dendrogram, while the QTP region and the Tethyan region were clustered together [5], which is consistent with the results of our study (northwestern and southeastern clusters) (Figure 4a). Based on the research conducted by Ye et al. (2019) regarding the phylogenetics of angiosperms in China, it has been observed that the phytogeographic composition of Gansu Province is precisely situated at the confluence of four major phytogeographic realms within China, namely the Holarctic, the East Asian, the Tethyan, and the QTP (Fig 5). Specifically, (1) regions I, VII, and VIII belong to the QTP floristic region, with region I located at the northern edge of QTP and regions VII and VIII situated on the eastern edge of QTP; (2) region II belongs to the Tethyan floristic region, with rich components of the Tethyan flora that evolved differently from those of the Holarctic and East Asian regions but have a closer relationship to the QTP flora [5]; (3) regions III-V belong to the Holarctic flora region, with rich components of the Holarctic Tertiary flora and a close relationship to the East Asian floristic region [5]; (4) region â…¨ belong to the East Asian floristic region, preserving the once widely distributed North Tropical flora of the Northern Hemisphere that became extinct in other areas due to the influence of the Quaternary ice age [88-90].

The intricate nexus of phytogeographic regiens in Gansu Province assumes profound significance [47]. The amalgamation of these regions engenders a heterogeneity of biotic elements, resulting in a region characterized by an elevated richness and intermingling of plant taxa [91]. This conspicuous convergence brings forth the juxtaposition of diverse flora, each with its own historical trajectory, ecological adaptation, and evolutionary trajectory, thereby catalyzing a mosaic of ecological niches and selective pressures that underpin the observed botanical diversity [92]. In summation, the positioning of Gansu Province as an intricate phytogeographic nexus bears profound implications for botanical diversity. Its status as an intersection of realms amplifies the significance of this region in the discourse of plant evolution, biogeography, and ecological adaptation. The intricate interplay of these realms within Gansu Province warrants sustained scholarly attention, promising a deeper comprehension of the underlying drivers and consequences of this unique botanical confluence.

Q8: Fig. 3, it is not clear how the expected estimates were obtained, clarify

Response: We sincerely thank the reviewer for careful reading. Within this part, it is important to note that the standardized effect size of pβsim (SES.pβsim) carries a distinct meaning from pβsim itself. Because pβsim is likely to be related to βsim, we used a null model to test if county-level assemblages were more or less phylogenetically similar than expected based on the number of taxa. Null distributions were employed to calculate the standardized effect sizes of pβsim (SES. pβsim), where the mean of the null distribution was subtracted from the observed pβsim and divided by the standard deviation of the null distribution. A positive SES signifies a higher degree of phylogenetic turnover than that expected from a random sampling of the regional flora, whereas a negative SES.pβsim denotes the opposite. However, the Pβsim value is the proportion of shared phylogenetic branch lengths of the dated phylogenetic tree between regions. Pβsim ranges from 0 (when genera are identical and share the same branch lengths) to 1 (when genera do not share similar branch lengths). In the Materials and Methods section, we explained in detail the ecological significance of the estimates, the calculation formula and the scientific meaning represented by the size of the results. In order to make our article better understood, we have refined it according to your comments. As follows: (Lines 209-245). Methodology comes from Baselga et al. (2012).

To quantitatively evaluate the spatial turnover of floristic composition in Gansu, we utilized the turnover component of the Sørensen dissimilarity index (βsim). The βsim index allows us to assess the taxonomic turnover that occurred between each pair of regions. The calculation of βsim (1) is performed as follows:

                              (1)

Where 'a' represents the count of shared genera between two regions, while 'b' and 'c' correspond to the counts of genera that are unique to each respective region. The values of βsim range from 0 to 1, with 0 indicating an identical generic composition between the regions, and 1 denoting the absence of shared taxa between the regions. One notable advantage of the βsim metric is its ability to mitigate the impact of species richness heterogeneity across different regions, thus ensuring a robust assessment of compositional turnover.

To quantify the phylogenetic turnover in our study, we utilized phylogenetic beta diversity (pβsim), an adapted version of the βsim index. Instead of considering the proportion of shared genera, pβsim assesses the proportion of shared phylogenetic branch lengths in the dated phylogenetic tree between regions. The values of pβsim range from 0 to 1, where 0 indicates that the genera have identical branch lengths and share the same evolutionary history, while 1 indicates that the genera have distinct branch lengths and do not share similar evolutionary trajectories. Pairwise distance matrices were calculated for both pβsim and βsim across all regions. Additionally, we compared the obtained values of βsim and pβsim with those derived from the Jaccard Index (βjaccard) to assess the sensitivity of our results (βjtu and pβjtu).

To investigate the potential association between pβsim and βsim, we employed a null model approach to assess whether county-level assemblages exhibited greater or lesser phylogenetic similarity compared to what would be expected based solely on the number of taxa. This null model analysis involved generating a null distribution by randomly permuting the tips of the phylogenetic tree a total of 999 times. For each permutation, pairwise distance matrices were computed for both pβsim and βsim, capturing the phylogenetic turnover between all regions while preserving the inherent taxonomic turnover differences among them. Subsequently, pβsim values were calculated for each pair of compared regions by generating pairwise distance matrices for pβsim. Standardized effect sizes of pβsim (SES.pβsim) (2) were obtained by comparing the observed pβsim value to the mean of the null distribution and dividing it by the standard deviation of the null distribution. Positive values of SES.pβsim indicate a higher level of phylogenetic turnover than expected under random sampling, while negative values indicate the opposite scenario.

                      (2)

Q9: Lines 202-205, Why would not define just two biogeographic regions (I and II) and III to IX, this has to be included as an alternative within the discussion section

Response: We sincerely thank the reviewer for careful reading. From the perspective of the degree of change in the phylogenetic composition of cross-species combinations in Gansu Province based on hierarchical clustering of phylogenetic β diversity (pβ) values, the results generated established a total of nine floras nested in two larger fields (Fig.4) (Supplementary Fig S3, S4). The results of both NMDS analysis and cluster analysis indicated that our division of the flora is feasible and reliable (Line 328-335 and lines 366-370). According to reviewer's comment, we have refined this issue within the discussion section. As follow: (Lines 439-447)

Our research not only captured the two broader phytogeographic differentiations among plant assemblages within Gansu Province, but also embraced the subtler ecological nuances inherent in its botanical composition. The incorporation of nine subordinate subregions within the overarching biogeographic classification serves to underscore the intrinsic ecological intricacy permeating Gansu Province. We delineated Gansu Province into multiple levels of biogeographic divisions, encompassing both broader and more localized scales, emanates from our study aspires to provide a holistic understanding that reflects the true nature of the province's biotic diversity and its historical biogeographic significance.

Q10: Lines 211-212, why would there by a high or low correlation between taxonomy and phylogeny?

Response: We sincerely thank the reviewer for careful reading. The correlation between taxonomy and phylogeny, specifically the degree to which they align or diverge, is influenced by a complex interplay of biological, methodological, and conceptual factors. The correlation can vary from high to low due to a combination of the following reasons:

  1. Taxonomy is based on species classification, often reflecting historical descriptions and morphological characteristics. As our understanding of evolutionary relationships improves through molecular data and advanced analytical methods, taxonomic revisions may lead to discrepancies between the traditional classification and the newly inferred phylogenetic relationships.
  2. Convergent evolution occurs when distantly related species independently evolve similar traits due to shared environmental pressures. Such cases can result in species with distinct phylogenetic origins being grouped together taxonomically, leading to a low correlation between taxonomy and phylogeny.
  3. Polyphyletic groups consist of species that do not share a common ancestor within the group. If such groups are recognized taxonomically, they would result in low correlation, as the taxonomic group does not reflect true evolutionary relationships.
  4. If a taxonomic group is incompletely sampled or not fully represented in phylogenetic analyses, the resulting correlation between taxonomy and phylogeny might be low due to missing evolutionary information.
  5. Taxonomic lumping (combining species) or splitting (dividing species) decisions might not always align with phylogenetic relationships. Taxonomists may prioritize other criteria, such as practicality, morphological similarities, or ecological considerations, which could lead to discrepancies.
  6. Cryptic species are morphologically similar but genetically distinct. If cryptic species are treated as a single taxonomic entity, the correlation between taxonomy and phylogeny would be low, as phylogenetic relationships are not accurately reflected in the taxonomy.
  7. Hybridization events between species can blur phylogenetic relationships and lead to challenges in taxonomic classification. Hybrids might possess genetic material from multiple lineages, causing taxonomic and phylogenetic discordance.
  8. Phylogenetic trees might lack full resolution due to limited genetic data or rapid radiation events. This can lead to a mismatch between taxonomy and phylogeny, as relationships are not fully captured.
  9. The choice of phylogenetic methods, molecular markers, and data analysis techniques can influence the resulting tree topology, affecting the correlation with taxonomy.
  10. Different genes or regions of the genome can evolve at varying rates. This can lead to incongruences in the inferred phylogenetic relationships compared to taxonomic groupings.

In summary, the correlation between taxonomy and phylogeny can be high when traditional classifications accurately reflect evolutionary relationships, especially when comprehensive molecular data and advanced methods are employed. However, factors like convergent evolution, taxonomic revisions, hybridization, and incomplete sampling can lead to low correlations. Addressing these complexities requires a multidisciplinary approach that integrates morphological, molecular, ecological, and historical data to achieve a comprehensive understanding of the relationships between taxa.

Q11: Lines 214-216, this implication appears already known and published and it can be deleted

Response: According to reviewer's comment, we have deleted it.

Q12: Lines 257-355, why is it necessary to define phylogenetically floristic regions if the taxonomically defined are so similar, please justify

Response: According to reviewer's comment, we have refined the figure. As follow: Defining phylogenetically floristic regions, even when taxonomically defined regions show similarities, can provide valuable insights into the evolutionary history, ecological processes, and conservation priorities of a biogeographic area.

Phylogenetic relationships provide a window into the historical processes that have shaped species distributions. Even if taxonomically defined regions are similar, phylogenetic data might reveal historical connections and divergence events that are not apparent from morphological or taxonomic characteristics alone. Phylogenetic analyses can uncover ancient lineages and cryptic species that are not readily distinguishable using traditional taxonomy. These unique evolutionary histories might be obscured when relying solely on taxonomic similarity, potentially leading to underestimations of diversity and evolutionary significance. Phylogenetic regionalization can identify areas that have served as evolutionary hotspots or refugia, contributing to the preservation of lineage diversity. These areas might not be apparent in taxonomic analyses alone and can guide conservation efforts by protecting key evolutionary processes.

Phylogenetic relationships often correlate with ecological traits and adaptations. Identifying phylogenetic regions can help reveal ecological patterns associated with specific lineages, aiding in understanding ecosystem function and adaptation to local conditions. Predicting how species distributions might change in response to environmental shifts relies on understanding historical biogeography. Phylogenetic data can provide insights into how lineages responded to past climatic changes and can inform projections for the future. Phylogenetic regions might reveal areas of high endemism and unique evolutionary trajectories. Incorporating phylogenetic information into conservation planning ensures that evolutionary diversity is protected alongside taxonomic diversity.

Taxonomy is subject to revisions as our understanding of species relationships evolves. Phylogenetic information can guide taxonomic decisions, reducing the risk of misclassifying species due to morphological similarities. Phylogenetic regionalization allows for testing hypotheses about historical biogeographic events, dispersal patterns, and evolutionary processes. Such insights can enhance our understanding of biogeographic mechanisms.

Q13: Lines 350-355, this paragraph is too general, it could be deleted

Response: According to reviewer's comment, we have deleted it.

Q14: Lines 490-491, note that while the approach is novel, the results change very little the biogeography of the flora that has been already defined, include this in the conclusions.

Response: We sincerely thank the reviewer for comments. we have refined this issue within this section. As follow: (Lines 551-563)

In this study, a novel approach utilizing phylogenetic relationships has been employed to delineate the floristic regions within Gansu. This approach provided a more comprehensive understanding of the biogeographic patterns and evolutionary history of plant lineages within the region, by incorporating information on the shared ancestry and diversification history of taxa. Compared with the traditional method which only depends on species distribution, it provides a more detailed perspective for the formation and maintenance of flora assemblages, and its flora results also show a slight difference from the traditional method.

Q15: Lines 499-501, this assertion is obvious and it could be deleted.

Response: According to reviewer's comment, we have deleted this assertion.

Q16: Lines 505-533, this paragraph belongs to the discussion section and also the ideas should be included and properly referenced in the introduction

Response: According to reviewer's comment, we have refined this issue and referenced in the introduction. As follow: (Lines 141-149 and Lines 535-549)

Phytogeographic regionalization holds crucial significance in comprehending the evolutionary dimension of floristic diversity, particularly in the specified area [52]. By investigating the distribution patterns and evolutionary histories of plant species, it offers insights into conservation strategies, biogeographic evolution, ecosystem functioning, and taxonomic research [53]. To reinforce regionalizations, integrating data from various species groups helps unveil co-evolutionary relationships, predict ecosystem shifts, and enhance our understanding of the region's history and potential future changes [54]. This approach not only aids in delineating phylogenetically informed floristic regions but also deepens our grasp of phytogeographical establishment and identification [55].

From an evolutionary perspective, this study's findings open new avenues for exploring floristic assemblages in Gansu province: 1). Investigating historical biogeographic processes that shaped plant diversity and distribution patterns by tracing lineage evolution and geographic spread can reveal key factors driving current trends. 2). Future studies can uncover how phylogenetic relationships influence community assembly, including the roles of ecological filters, competition, and coexistence mechanisms. This sheds light on the evolutionary dynamics of local plant communities. 3). Research into how specific environmental factors, like climate and topography, influence lineage diversification in Gansu offers insights into the interplay between ecology and evolution. 4). Comparing Gansu with diverse regions can elucidate general patterns, unique features, and the relative impact of historical processes and contemporary factors on plant diversity. 5). Utilizing phylogenetic regions for conservation planning enhances the protection of areas with high phylogenetic diversity, endemism, and evolutionary distinctiveness, preserving both species and evolutionary history.

Reviewer 2 Report

This manuscript entitled "Phylogenetic Partitioning of Gansu Flora: Unveiling the Core Transitional Zone of Chinese Flora" proposes an interesting methodology to establish biogeographical models using a phylogenetic approach. The paper complies with the editorial line of the journal. The results are consistent with the methodology employed. The methodology used is adequate in relation to the initial hypotheses and the objectives of the work.

However, from a conceptual point of view, the paper raises some important doubts.

The first is that it is based and supported by a phylogenetic model for botanical genera that is still far from robust for many genera. This is largely because phylogenetic classifications of higher plants (APG) are strictly molecular, ignoring phylogeny and evolutionary history from other approaches.

This has repercussions on the authors' results, which, in the event of a change of phylogenetic model, their results would vary, making the results of the present manuscript not very robust.

Another important shortcoming lies in the division of Gansu province on the basis of non-natural criteria such as an administrative division, which can make the results misleading and difficult to interpret or extrapolate to other studies as it is an arbitrary division.A choice of areas of presence/absence by stratified sampling would solve this problem, taking into account bioclimatic, edaphic, geological, topographical or phytosociological criteria.

However, more detailed comments are attached as marginal comments in the manuscript.

The manuscript has several spelling mistakes, as well as a lack of formatting and style.

Author Response

Dear Editor and dear reviewers,

Re: Manuscript ID: “plants-2503187” and Title: “Phylogenetic Partitioning of Gansu Flora: Unveiling the Core Transitional Zone of Chinese Flora”

Thank you for your letter and the reviewers’ comments concerning our manuscript entitled “Phylogenetic Partitioning of Gansu Flora: Unveiling the Core Transitional Zone of Chinese Flora” (ID: plants-2503187). Those comments are valuable and very helpful. We have read through comments carefully and have made corrections. Based on the instructions provided in your letter, we uploaded the file of the revised manuscript. Revisions in the text are shown using red highlight for additions, and strikethrough font for deletions. The responses to the reviewer's comments are marked in blue and presented following.

We would love to thank you for allowing us to resubmit a revised copy of the manuscript and we highly appreciate your time and consideration.

Sincerely.

Zizhen Li.

Reviewer #2:

Q 1: Line 11-34, Summary too broad. Many aspects do not need to be advanced in the summary, as they will be developed later.

Response: We sincerely thank the reviewer for careful reading. We have made modifications to the wording of this sentence in our manuscript (Lines 11-30). As follow:

Floristic regions, conventionally established through species distribution patterns, have often overlooked the phylogenetic relationships among taxa. However, how phylogenetic relationships influence the historical interconnections within and among biogeographic regions remains inadequately understood. In this research, we compiled distribution data of seed plants in Gansu, a biogeographic diversity region located in northwestern China. We propose a system of floristic regions within Gansu by combining data on the distributions and phylogenetic relationships of genera-level of native seed plants, aiming to explore the relationship between phylogenetic relatedness, taxonomic composition, and regional phylogenetic delineation. We found that 1) phylogenetic relatedness was strongly correlated with the taxonomic composition between floras in Gansu. 2) The southeastern Gansu region shows the lowest level of spatial turnover in both phylogenetic relationships and taxonomic composition of the floristic assemblages across the Gansu region. 3) Null model analyses indicated nonrandom phylogenetic structure across the region, where most areas showed higher phylogenetic turnover than expected given the underlying taxonomic composition between sites. 4) Our results demonstrated a consistent pattern across various regionalization schemes and highlighted the preference for employing the phylogenetic dissimilarity approach in biogeographical regionalization investigations. 5) Employing the phylogenetic dissimilarity approach, we identified nine distinct floristic regions in Gansu, categorized into two broader geographical units, namely the northwest and southeast. 6) Based on the phylogenetic delineation, we obtained explicit ranges and intersection boundaries for the four major phytogeographic regions of China across this area.

Q2: Line 36, To improve the visibility of your publication, please try not to repeat the words in the title of the manuscript in the keywords.

Response: We sincerely thank the reviewer for careful reading. We have corrected this mistake in the revised manuscript. As follow: (Lines 31-32).

Keywords: floristic regions, phylogenetic relationships, phylogenetic beta diversity, spatial turnover, seed plants.

Q3: Line 38. It would be interesting to mention some model of biogeographical classification, as well as the biogeographical framing of the Gansu area,

I recommend reading other literature such as: Rivas-Martínez et al. (2011): Biogeographic Map of South America. A preliminary survey. Rivas-Martínez et al. (2017): Biogeographic units of the Iberian Peninsula and Baelaric Islands to district level. A concise synopsis Mucina (2023): Synthesis: A New Global Zonobiome Paradigm

Rivas-Martínez & Penas (2011): Rivas-Martínez S, Sáenz SR, Penas A (2011) Worldwide bioclimatic classifcation system.

On the other hand, any biogeographical approach to an area must imply an approximate knowledge of the different plant communities that make up that area. The introduction does not mention such important aspects as plant communities, or the possible phytosociological relationships that may exist.

Response: We are grateful for the suggestion. As suggested by the reviewer, we have cited some phytogeographical frameworks of the flora that are more in line with our research content, which is more theoretical support for our research content. And we also added more details of plant communities in Gansu. As follow: (Lines 43-47 and Lines 115-140)

Several biomes and biogeographical regionalization schemes have been proposed, such as the world’s flora, East Asian Plants, the flora of China, and the biogeographical framing of the Gansu area. These schemes offer a spatially explicit framework to investigate biodiversity patterns on a large scale.

Gansu province is characterized by a complex physical geography, a rich biodiversity, and an ancient floristic component with many endemic taxa. It encompasses four plant subregions: Holarctic flora, East Asian flora, Tethyan flora, and Qinghai-Tibet Plateau flora and has seven geomorphic types: mountains, plains, basins, plateaus, valleys, deserts, and gobi. Due to its unique characteristics, Gansu province serves as a pivotal intersection and transitional zone for the geographical distribution of numerous plant communities, not only within China but also across Asia. The region showcases a rich variety of vegetation types, each adapted to its specific ecological niche. From lush alpine meadows to arid desert oases, Gansu's plant communities span a wide range of altitudes, climates, and soil conditions. This diversity is shaped by the complex interplay of geographical features, including the Qilian Mountains, the Hexi Corridor, and the Yellow River. These various plant communities play a crucial role in maintaining local and regional biodiversity. Early studies on the flora of Gansu have been predominantly limited to taxonomic approaches, focusing on particular plant taxa or geographical regions. And previous studies in plant phylogenetics have predominantly concentrated on geographically homogeneous hotspots, such as North America, Southern Africa, the Lesser Antillean of Central America, Yunnan of China. However, exploring the differences of plant diversity in the evolutionary dimension and understanding the distribution boundaries of different plant regions in Gansu, with complex geographical environments, diverse climates and significant differences in vegetation, is extremely attractive. No prior investigation has integrated both phylogenetic and spatial aspects to explore the evolutionary trajectory of the Gansu flora. We lack an understanding of the critical evolutionary dimension of floristic diversity in this area. Investigating the phytogeographic regionalization of this area is of great significance and value in enhancing our understanding of the differentiation and historical progression in transition zones of Chinese and even Asian floras.

Q4: Line 50-52, The possible phylogenetic relationships between the different taxa cannot be understood without a formal bioclimatic and palaeobiogeographic contextualisation. It would be interesting to add bibliography in this respect.

Response: Thank you for your suggestion. As suggested by reviewer, we have added the suggested content to the manuscript as follow: (Line 47-52)

Nonetheless, a notable constraint observed in preceding studies is their failure to incorporate a deliberate examination of the phylogenetic interconnections among species. This oversight has led to an inadequacy in portraying the multifaceted evolutionary chronicle of floristic territories, as well as a deficiency in bioclimatic and palaeobiogeographic contextual elucidation.

Q5: Line 55, Please clarify in what sense the authors mean when they use the term "phylogenetic", do they refer to a molecular aspect, or do they mean another type of approach?

Response: Thank you for your comments, the discussion regarding this question is presented following:

"Phylogenetic " pertains to the quantitative characterization of evolutionary relationships and bifurcation patterns among taxa within a designated biological cohort. This construct encapsulates the range of lineages, genetic affinities, and evolutionary intervals encompassed within a taxonomic assemblage. By attending to not only the taxonomic profusion but also the underlying evolutionary lineage that interconnects the constituents, this concept provides a nuanced vantage point for comprehending biodiversity, unveiling the intricate mosaic of evolutionary mechanisms that have shaped a specific biological composition.

The computation of phylogenetic diversity commonly entails the construction of a phylogenetic tree, or cladogram, which visually represents the evolutionary interconnections among taxa. Rooted in the genetic data of the species, such as DNA sequences, the phylogenetic tree arranges taxa in a manner that reflects their historical divergence and evolutionary relationships. Within this tree, more closely related species cluster together, while more distantly related ones occupy more dispersed positions.

Quantifying phylogenetic diversity often involves the derivation of a phylogenetic diversity index, determined by considering the branch lengths and bifurcation patterns among taxa in the phylogenetic tree. Longer branches signify greater evolutionary divergence; hence, in the computation of this index, greater branch lengths are accorded higher phylogenetic diversity.

It is important to note that the calculation of phylogenetic diversity may vary depending on specific research objectives and methodologies. However, as a whole, it furnishes an approach to appraise biodiversity that incorporates evolutionary relationships, thereby enriching our holistic understanding of biotic diversity within ecosystems.

The specific method is in the manuscript lines 209-220. Methodology comes from Baselga et al (2012).

Q6: Lines 62, It would be interesting to clarify the concept of phylogenetic structure of a regional plant assemblage.

Response: Thank you for your suggestion. As suggested by reviewer, we have added the suggested content to the manuscript as follow: (Line 61-67)

The phylogenetic structure of a regional plant assemblage refers to the arrangement and distribution of plant species within a specific geographic area based on their evolutionary relationships. Qian et al. (2013), Li et al. (2014), and Swenson and Umaña (2014) revealed that the phylogenetic structure of regional plant assemblages is determined by environmental conditions and biogeographical history, from the latitudinal gradient, latitudinal diversity gradient, and environmental heterogeneity.
Q7: Line 66-70, Although referencing Mishler et al. and other published works, please briefly describe the concepts in a couple of lines. This helps to contextualise and better understand the manuscript.

Response: Thank you for your suggestion. As suggested by reviewer, we have added the suggested content to the manuscript as follow: (Line 67-76)

Phylogenetic Diversity (PD) refers to the measurement of the evolutionary relatedness and diversity of species within a biological community or ecosystem. Phylogenetic Endemism (PE) pertains to the concentration of unique or endemic lineages in a specific geographic region. Relative Phylogenetic Diversity (RPD) is a standardized measure that compares the observed phylogenetic diversity of a particular site or community to the phylogenetic diversity expected under a null model, often accounting for factors such as species richness. Relative Phylogenetic Endemism (RPE) similar to RPD, RPE is a relative measure that assesses the concentration of phylogenetically unique species within a specific area compared to a null model.
Q8: Line 76-78, What are these evolutionary processes, and how can these evolutionary processes be interpolated in the context of the study of flora?

For example, the quote [33]: Moritz 2002, deals with the various phylogeographic factors in the fauna of Australia but not in the flora. While emphasising migration gradients, this concept is hardly applicable to flora, since other mechanisms of dispersal and adaptation are involved.

Response: Thank you for your comments, the discussion regarding this question is presented following:

The evolutionary processes that maintain and generate biodiversity are fundamental mechanisms that contribute to the variety of life forms and their interactions in ecosystems. These processes can be categorized into several key aspects:

  1. Mutations introduce genetic diversity by altering the DNA sequences of organisms. This variation serves as raw material for natural selection and adaptation to changing environments.
  2. Organisms possessing advantageous traits that enhance their survival and reproduction are more likely to pass on their genes to the next generation. Over time, these beneficial traits become more prevalent within a population.
  3. In response to new ecological niches or opportunities, a single ancestral species can give rise to multiple descendant species with diverse adaptations. This process often leads to rapid speciation and increased biodiversity.
  4. Migration and gene exchange between populations can introduce new genetic material, promoting diversity within and among populations.
  5. Random events can cause certain genetic variants to become more or less common within a population. In small populations, genetic drift can lead to significant changes in allele frequencies over time.
  6. Interspecies mating can lead to hybrid offspring with unique genetic combinations, contributing to genetic diversity and potential adaptation.
  7. Species that interact closely, such as predator-prey or host-parasite relationships, can drive reciprocal adaptations between species, leading to increased diversity in both.
  8. Competition, predation, mutualism, and other ecological interactions can shape the characteristics of species and their adaptations within an ecosystem.

In the context of the study of flora, these evolutionary processes can be interpolated by examining the phylogenetic relationships among plant species, their adaptations to different environments, and their distribution patterns. By analyzing the genetic variation, functional traits, and ecological interactions of plant species, researchers can infer how these evolutionary processes have led to the diversity of plant life in a specific region.

According to the reviewer 's opinion, we modified the citation “Winter, M., Devictor, V., Schweiger, O. (2013). Phylogenetic diversity and nature conservation: where are we? Trends in Ecology & Evolution, 28, 199-204”.

Q9: Line 107-116, Please separate this paragraph from the previous one. Here the authors set out the starting hypothesis and part of the objectives.

Response: We sincerely thank the reviewer for careful reading. We have corrected this in the revised manuscript (Line 132-140).

Q10: Line 131, the Material and methods section is best viewed here.

Every time an acronym appears for the first time, it has to be explained.

Response: We sincerely thank the reviewer for careful reading. We have corrected this in the revised manuscript (Line 164-296).

Q11: Line 134, Where is figure 1? In the text, figure 2 appears before figure 1.

Why did you choose an administrative division of Gansu province and not a more natural bioclimatic division?

Response: We sincerely thank the reviewer for careful reading. We have corrected this mistake in the revised manuscript.

The decision to use an administrative division of the territory was driven by:

  1. Administrative geographical unit and natural geographical unit are highly overlapped.
  2. Data are mainly recorded in the form of administrative geographic units.

Q12: Lines 164-169, This paragraph is confusing and does not fully clarify why the authors decided to use pβsim to establish the different phytogeographical regions. From what data have the authors inferred this decision?

Rewrite this expression, in your work you do not talk about the vegetation of Gansu (vegetation being understood as the set of plant communities characterised in a territory). Therefore, your work talks about the relationships between the different taxa, not about vegetation.

Response: We sincerely thank the reviewer for careful reading. The pβsim value has been routinely used in the calculation of phylogenetic diversity. Since Graham et al. (2018) proposed this theory, it has been used in the calculation of this research field. It is a conventional and representative calculation method, which has been used by many scholars in recent years, such as Li et al. (2015), Ye et al. (2019), Daru et al. (2016).

We have modified the improper use of words you mentioned(Line 333).

Q13: Lines 279, What could account for this discrepancy?

Response: Thank you for your comments, It is caused by two different theories and division methods.

Phylogenetic dissimilarity refers to the divergence in evolutionary relationships among species within different geographical regions or ecological communities. Taxonomic dissimilarity, on the other hand, reflects the degree of difference in species composition based on their classification into taxonomic groups. The extent of association between these two dissimilarities is influenced by several factors:

  1. Species within the same taxonomic group share a common evolutionary history to varying degrees. Taxonomic dissimilarity patterns might be driven by phylogenetic relationships; closely related species are more likely to be categorized within the same taxonomic group. However, exceptions arise due to factors like convergent evolution, hybridization, and taxonomic revisions.
  2. Ecological processes, such as adaptation to local environments, could result in similar species being present across different regions, leading to lower taxonomic dissimilarity despite considerable phylogenetic divergence. Conversely, closely related species might occupy distinct niches, contributing to high taxonomic dissimilarity and moderate phylogenetic dissimilarity.

3.Historical events like dispersal, vicariance, and extinction shape species distributions. Phylogenetic dissimilarity could reflect historical biogeographic events more directly, whereas taxonomic dissimilarity might be influenced by factors such as dispersal abilities and colonization history.

  1. Taxonomic classifications might not always accurately represent evolutionary relationships. Taxonomic revisions, lumping, or splitting of taxa can influence taxonomic dissimilarity patterns without necessarily impacting phylogenetic dissimilarity in the same way.
  2. Sampling intensity and spatial scale can affect observed patterns. Uneven species representation or scale differences might lead to inconsistencies between phylogenetic and taxonomic dissimilarities.

Theoretical associations between these dissimilarities can vary:

  1. In cases where evolutionary history closely aligns with taxonomic classification, phylogenetic and taxonomic dissimilarity patterns might be congruent. Regions with high phylogenetic dissimilarity could exhibit high taxonomic dissimilarity due to the accurate representation of evolutionary divergence in taxonomy.
  2. Phylogenetic and taxonomic dissimilarities might diverge due to ecological factors, historical biogeography, or taxonomic uncertainties. High phylogenetic dissimilarity could coincide with low taxonomic dissimilarity if niche-driven ecological processes lead to non-random species composition.
  3. Various scenarios can exist where partial congruence or discordance exists between the two dissimilarities, illustrating the complex interplay between evolutionary and taxonomic factors.

Ultimately, understanding the theoretical associations between phylogenetic and taxonomic dissimilarity patterns requires a holistic consideration of evolutionary, ecological, and historical dynamics. Empirical studies, incorporating robust phylogenetic reconstructions and comprehensive taxonomic assessments, are crucial to untangle the intricate relationship between these dissimilarities within the Gansu flora and other biological systems.

Q14: Line 283, What is the meaning of "plant group" in this context?

Response: "Plant group" refers to a collection or category of plants that share common characteristics, traits, or attributes. It is a broader term used to describe a set of plant species that exhibit similarities in terms of their morphology, genetics, ecological preferences, or evolutionary relationships. These similarities could be at different levels of classification, ranging from a group of closely related species within a genus to a larger grouping such as a family, order, or even a higher taxonomic category.

Plant groups are often used to facilitate the study, categorization, and understanding of plant diversity. They help researchers, botanists, and ecologists classify and make sense of the vast array of plant species based on shared features and evolutionary relationships. Examples of plant groups include conifers, grasses, orchids, and dicotyledons, each of which comprises multiple species that exhibit certain common traits or relationships.

Q15: Line 305-311, Please check the text format.

Response: We sincerely thank the reviewer for careful reading. We have corrected this mistake in the revised manuscript.

Q16: Lines 323-328, The authors cannot infer from their results this discussion, or at least provide a reference. Furthermore, the precepts of reliability and robustness of the results presented and defended in this manuscript are based on a phylogenetic model for the different floristic taxa, in this case at the genus level. This model is still far from robust for many floristic genera of uncertain position, as the phylogeny is only considered from a strictly molecular point of view, without addressing other phylogenetic and evolutionary aspects such as what plant communities are composed of or their phytosociological, biocliomatic, or geographical and topographical aspects in general.

How do the authors explain that from artificial units such as the different administrative units of the province of Gansu, optimal regions can be extracted from a biogeographical point of view, since these regions are more natural?

Response: Thank you for your comments, the discussion regarding this question is presented following:

Phylogenetic divisions based on family-level phylogeny may be helpful for rough or preliminary phylogenetic divisions, but more detailed analysis requires phylogenetic divisions at the genus or species level, especially for plants with complex evolutionary history. The choice of taxonomic units in ecological and evolutionary analyses, such as those conducted in the manuscript, often depends on the specific research question, available data, and scientific rationale. In this case, the manuscript departs from the convention of using species as taxonomic units and instead employs genera for its analyses. The use of genera as taxonomic units could be justified if it aligns with the goals of the study. Genera might offer a balance between species-level resolution and higher-level taxonomic grouping, providing a compromise that captures both the diversity within and between species. This can be particularly relevant when species-level identification is challenging due to limited data availability, taxonomic uncertainties, or hybridization events. By analyzing at the genus level, the study aim to uncover broader patterns in ecological and evolutionary processes while avoiding potential noise or ambiguity associated with species-level identification. According to the reviewer’s comment, we have added a more detailed interpretation regarding in lines 189-196. As follows:

To analyze the spatial changes in species composition and phylogenetic relatedness, the study area was divided into 80 county-level geographical units (Supplementary Figures S1, Supplementary Table S1). Due to limitations in available data, taxonomic uncertainties, and hybridization events, species-level identification poses challenges. Hence, we decided to conduct the analysis at the genus level, in accordance with Li et al (2015) and Ye et al (2019). The presence or absence of each genus of seed plants in each county was ascertained by extracting distribution information of each genus and constructing a presence-absence matrix (Supplementary data matrix).

From the outcomes obtained in our study, it is evident that the data resolution at the genus level adequately supports the principal objectives and contents of our research. Furthermore, the results align with established patterns and theoretical expectations, thus affirming the feasibility of conducting our research at the genus level. The forward-looking issues you raised are also under consideration within the scope of our ongoing efforts. In order to make this difference more clear, we intend to undertake a more in-depth research by combining species-level data for Gansu endemic plants with environmental factors such as precipitation, mean temperature, and terrain variability.

While administrative boundaries may appear artificial of Gansu, they highly overlap due to historical factors and their interaction with the environment.

Q17: Line 359-360. Check the formatting of coordinates and character spacing.

Figure 2 appears in the text before figure 1, please check these figures.

Response: We were really sorry for our careless mistakes. We have corrected this mistake in the revised manuscript.

Q18: Line 362-364. Which climate model have you used?

In reference [72],

Temperature change characteristics in Gansu Province of China. Atmosphere 13, 728. The authors do not specify which climate model they have used. You have probably based yourselves on figure 1 of this publication. It would therefore be useful to specify which climate or bioclimatic model you have used for this research work.

I recommend a bioclimatic approach with an accurate global model such as the Rivas-Martínez World Bioclimatic classification: Rivas-Martínez, S., Rivas-Saenz, S., & Penas, A. (2002). Worldwide bioclimatic classification system.

Response:We sincerely thank the reviewer for careful reading. we have added the suggested content to the manuscript as follow (Lines 170-172).

It also spans four major climatic types: Warm Temperate Continental, Cold Temperate Continental, base on the Rivas-Martínez World Bioclimatic classification [57].

Q19: Line 367. Check spelling

Response: We were really sorry for our careless mistakes. This name is a policy put forward by the Chinese government. For people in other countries in the world may not know what it means, in order to reduce unnecessary misunderstandings, we deleted this passage.

Q20: Line 369.It would be interesting to talk a bit about the geology of the region with the main geological materials that can be found.

Response: Thank you for the suggestion. We have added the information required as explained above (Lines 116-119). As follow:

It encompasses four plant subregions: Holarctic flora, East Asian flora, Tethyan flora, and Qinghai-Tibet Plateau flora and has seven geomorphic types: mountains, plains, basins, plateaus, valleys, deserts, and gobi.

Q21: Lines 376-378. Please explain or elaborate on what is meant by "The species distribution data were diligently compiled".

What is the time interval between the earliest and latest reports? Have the authors considered that the distribution of any taxa might have changed in that time period?

Response: The data sources include field survey data provided by domain experts and scholars, online databases (such as Species 2000 China, Catalogue of Life China, CoL China (http://especies.cn), Chinese Plant Flora (www.iplant.cn), Chinese Biodiversity (http://sciencereading.cn), Chinese National Herbarium (www.cfh.ac.cn), Chinese Plant Thematic Database (www.csdb.cn), eFloras, The Plant List (www.theplantlist.org), etc.), as well as monographs, journal articles, and theses (such as "Flora of Gansu," "Botanical Chronicle of Lanzhou," "Vegetation of Gansu," "Resource Background Investigation of Gansu Xinglong Mountain National Nature Reserve," "Woody Plant Flora of Ziwu Ridge," "Checklist of Vascular Plants in Hexi Region of Gansu," "Arboreal Flora of Gannan," "Plants of Baishuijiang National Nature Reserve in Gansu," etc.) (Supplementary Material).

Plant taxonomy, a branch of biology, primarily focuses on the classification, naming, relationships, and evolutionary history of plants. Its main components encompass several aspects: species description and nomenclature to ensure unique scientific names for each species; establishment of a hierarchical classification system including orders, families, genera, and species; determination of classification criteria and features through the study of morphological, anatomical, physiological, and ecological traits; investigation of genetic relationships and evolutionary history through molecular biology techniques, constructing phylogenetic trees; continuous revision and updating of the classification system to reflect real plant relationships; examination of plant geographic distribution to understand their origins and dispersion; development and maintenance of databases, herbaria, and tools for researchers; and applications in agriculture, pharmacology, ecology, and conservation, aiding the protection of endangered plant species and ecosystems.

The environment for plant growth is relatively stable without major environmental changes or external interference, and will not change much in a short period of time. The constantly updated report is a supplement and improvement to the existing plant distribution database. Therefore, the dimension of time will be weakened in plant taxonomy.

Q22: Line 380. Please check the magazine's decimal number format: 5,244 or 5.244?

Have the authors only worked with botanical genera, or do they also refer to species level?

Response: Thank you for the suggestion. We have corrected this mistake in the revised manuscript.

We are more focused on the study of a single species or genus, such as Clematis, Rosa, etc.

Q23: Line 385. Why did the authors choose a political division of the territory and not a division based on biogeographical, geological, bioclimatic, etc. criteria?

Response: We sincerely thank the reviewer for careful reading. The decision to use a political division of the territory rather than a division based on biogeographical, geological, bioclimatic, and other criteria was driven by:

  1. Administrative geographical unit and natural geographical unit are highly overlapped.
  2. Data are mainly recorded in the form of administrative geographic units.

Q24: Line 405. This formula does not correspond to Sorensen's Distance but to Simpson's Distance. The authors should in this step clarify and if necessary redo the calculations and explain why they have used this index of dissimilarity or distance. The differences between these indices (Jaccard, Sorensen-Dice and Simpson) lie in the importance given to each component; in the case of the Sorensen index, the shared species are of great importance, which is why it is multiplied by two. In the case of the Simpson index, it is an index used when there are very high differences between pairs of communities, so we subtract the weight by obtaining the minimum value between b and c.

Response: We sincerely thank the reviewer for careful reading. The βsim value has been routinely used in the calculation of phylogenetic diversity. Since Baselga. (2012) proposed this theory, it has been used in the calculation of this research field. It is a very representative calculation method, which has been used by many scholars in recent years, such as Li et al. (2015), Ye et al. (2019), Daru et al. (2016). Now, the ' betapart ' package of R software provides convenience for non-mathematical researchers. In mathematical calculation, the following two literatures gives us the greatest help:

  1. Baselga, A. (2012). The relationship between species replacement, dissimilarity derived from nestedness, and nestedness. Glob. Ecol. Biogeogr. 21, 1223-1232.
  2. Baselga, A., and Orme, C. D. L. (2012). Betapart: An R package for the study of beta diversity. Methods Ecol. Evol. 3, 808-812.

(Table from Baselga. 2012)

Q25: Lines 415-416. What does this sentence mean, explain the concept of shared branch. It would be interesting to provide a figure or diagram explaining this concept, since it is a theoretical concept based on a supposed phylogenetic model proposed by [77].

Response: We sincerely thank the reviewer for careful reading. the concept regarding this question is presented following (This is a summary of the article Graham, C. H., and Fine, P. V. A. (2008). Phylogenetic beta diversity: linking ecological and evolutionary processes across space in time, the details can be seen in the original text):

Phylobetadiversity can be calculated in different ways. Hardy & Senterre (2007) and Chave et al. 2007 extended additive partitioning methods developed using species data to evaluate phylogenetic diversity in terms of both alpha-diversity and beta-diversity components. Both of these methods calculate the divergence time (branch lengths) between each pair of taxa and sum this difference among all possible pairs and use this continuous variable in the Simpson index, a traditional index for calculating beta diversity. Classic metrics to measure similarity between communities, such as the Jaccard or Sorenson’s index also could be explored (Magurran 2004). These metrics are calculated as the ratio of shared species to total species. In phylogenetic terms, this metric could be calculated as the total branch length covered by shared species relative to the total branch length covered by all species in both communities (Ferrier et al. 2007; Bryant et al. in press). In addition, Phylocom (Software for the Analysis of Phylogenetic Community Structure and Character Evolution, with Phylomatic; Webb et al. 2007, 2008) provides metrics to measure phylogenetic distance between samples by calculating either the mean branch-length distance for all possible pairs of taxa in one sample to the other or the nearest neighbour distance among samples. Future simulation studies are required to compare the relative utility of each of these approaches and the influence of a variety of different phylogenetic patterns. For instance, a community that included species from a phylogeny that had both very old and young species could result in uneven branch lengths in a community phylogeny pool a in higher probabilities of finding phylogenetic clustering at local scales (Kraft et al. 2007).

Figure 1 (a) Two islands, separated by 100 km of open ocean. Each island has a small mountain range in its centre, creating a rain shadow. Red areas represent dry forest, blue areas represent wet forest. In each island, community samples of all palm trees have been conducted at four one-hectare sites, two in each habitat type, separated by 25 km. Island 1 has been sampled four times, site a and site b in dry forest, and site c and site d in wet forest. Island 2 has been sampled four times, site e and site f in dry forest, and site g and site h in wet forest. (b) Hypothetical phylogenies of the palms from Island 1 and Island 2, exploring different combinations of geographical and ecological (habitat) structure. For simplicity, all species are restricted to only one habitat type and only one island, so all five of the phylogenies exhibit some degree of dispersal limitation and niche conservatism at the level of species. Clade type 1 is primarily structured by geography, and secondarily structured by habitat. Clade type 2 is primarily structured by habitat and secondarily structured by geography. Both of these clades exhibit niche conservatism with respect to wet and dry forest, and dispersal limitation with respect to island. Clade type 3 is structured by geography but not habitat and exhibits niche lability and dispersal limitation. Clade type 4 is structured by habitat but not by geography and exhibits niche conservatism but no dispersal limitation. Clade type 5 exhibits random structure with respect to geography and habitat and is the basis for the null expectation for the phylobetadiversity graphs in part C. For each hypothetical phylogeny, see Table in part D to see what pattern of community phylogenetic structure would arise in each site, and the graphs in part C for phylobetadiversity measures among sites of similar habitat type and between habitats, within and between islands. (c) Phylobetadiversity measures among sites of similar habitat type (red lines indicate comparisons among dry forest sites, blue lines indicate comparisons among wet forest sites) and among divergent habitat types (purple dashed lines indicate comparisons among wet and dry forest sites). The x-axes represent spatial scale, and smaller values on this axis represent within island comparisons. The horizontal line in the graphs are the null expectation of phylobetadiversity among sites given no geographic or ecological structure (see clade type 5). Points below the line indicate lower than expected phylobetadiversity and points above the line indicate higher than expected phylobetadiversity. For this example, site comparisons are presented as either between habitat types or within habitat types, but this kind of analysis could also be undertaken comparing sites with low vs. high variance in a particular environmental variable (with respect to region-wide sampling). (d) A Table showing for each site, a–h, the habitat type and the species of palms sampled in each site. For example, site ‘a’ contains a random sample of all dry forest palms found in Island 1 (see part B). On the right, the table shows what the pattern of community phylogenetic structure within each site, depending on what clade type the palms belonged to (see part B). Big ’C’, strong phylogenetic clustering (the sample of species is more closely related than a random expectation); little ’c’, weak phylogenetic clustering; r / o, random patterns and / or phylogenetic overdispersion (the sample of species within a community is less closely related than a random expectation; see Webb et al. 2002). Note that for clade types 1–4, all community samples are predicted to exhibit phylogenetic clustering (Graham et al. (2008)).

(Graham et al. 2008)

Q26: Line 472. Why did they choose that number of random permutations?

Response: Thank you for your comments, the discussion regarding this question is presented following: 

The decision to employ 999 random permutations in the null model analysis is likely driven by several statistical considerations. One key rationale is to ensure statistical significance and robustness of results. Employing a high number of iterations, such as 999, is a common practice to mitigate the influence of random chance on the outcomes. This extensive permutation approach enhances the accuracy of estimating the null distribution and the likelihood of observing the observed data under the null hypothesis. Additionally, the choice strikes a balance between precision and computational efficiency, as a higher number of permutations demands more computational resources. Moreover, researchers often determine an optimal number of permutations through preliminary studies, aiming to stabilize the distribution without excessive computational burden. The adoption of 999 permutations could also reflect established conventions within the field or sensitivity analysis to validate the findings. Overall, the selection of 999 random permutations aligns with the pursuit of statistically meaningful and dependable outcomes, while accommodating computational limitations and adhering to customary practices.

Q27: Lines 494-495. This conclusion cannot be inferred from his results, as he has not compared a traditional species distribution approach with the approach proposed in this manuscript

Response: We are extremely grateful to reviewer for pointing out this problem. We have deleted it.

Q28: Lines 507-533. Please restructure the conclusions into paragraphs that are easy to read.

Response: Thank you for the suggestion. Combined with the Q16 of reviewer 1, we modified this passage and moved it to the discussion section. As follow: (Lines 535-549)

From an evolutionary perspective, this study's findings open new avenues for exploring floristic assemblages in Gansu province: 1). Investigating historical biogeographic processes that shaped plant diversity and distribution patterns by tracing lineage evolution and geographic spread can reveal key factors driving current trends. 2). Future studies can uncover how phylogenetic relationships influence community assembly, including the roles of ecological filters, competition, and coexistence mechanisms. This sheds light on the evolutionary dynamics of local plant communities. 3). Research into how specific environmental factors, like climate and topography, influence lineage diversification in Gansu offers insights into the interplay between ecology and evolution. 4). Comparing Gansu with diverse regions can elucidate general patterns, unique features, and the relative impact of historical processes and contemporary factors on plant diversity. 5). Utilizing phylogenetic regions for conservation planning enhances the protection of areas with high phylogenetic diversity, endemism, and evolutionary distinctiveness, preserving both species and evolutionary history.

Round 2

Reviewer 2 Report

First of all, I would like to thank the authors for their Herculean effort to address each and every one of the issues raised by the reviewers. The manuscript is now much, much better.

As far as this reviewer is concerned, there is still a doubt about the use of artificial or political delimitations for biogeographical approaches, although this approach is no longer within the remit of the review of the manuscript and the clarifications made by the authors are sufficient. 

On the other hand, the authors still do not clarify the formula for line 293, the authors refer to Baselga (2012), however, comparing both formulas, it can be seen that they call Sorensen's dissimilarity index βsim=1-(a/(min(b,c)+?)) while Baselga 2012, in table 1, calls Sorensen's index the following formula: (b+c)/(a+b+c). The authors, in line 293, call Sorensen's Dissimilarity a formula that Baselga (2012) calls Simpson's index (more precisely, it is a rotation of Sorensen's index). This reviewer sees the use of this index as appropriate and has no objection to it, but the authors contradict themselves. According to Baselga (2012), the formula given in line 293 belongs to the "Sorensen family of indices" but does not correspond to the Sorensen index. Baselga (2012) in table 1, specifies: 

Sorensen: βsor= (b+c)/(2a+b+c)

Simpson: βsim = min (b,c)/(a+min(b+c))

(this last formula corresponds to the one shown on line 293).

Author Response

Dear Editor and dear reviewers,

Re: Manuscript ID: “plants-2503187” and Title: “Phylogenetic Partitioning of Gansu Flora: Unveiling the Core Transitional Zone of Chinese Flora”

Thank you for your letter and the reviewers’ comments concerning our manuscript entitled “Phylogenetic Partitioning of Gansu Flora: Unveiling the Core Transitional Zone of Chinese Flora” (ID: plants-2503187). Those comments are valuable and very helpful. We have read through comments carefully and have made corrections. Based on the instructions provided in your letter, we uploaded the file of the revised manuscript. Revisions in the text are shown using red highlight for additions, and strikethrough font for deletions. The responses to the reviewer's comments are marked in blue and presented following.

We would love to thank you for allowing us to resubmit a revised copy of the manuscript and we highly appreciate your time and consideration.

Sincerely.

Zizhen Li.

Reviewer #2:

Q1: As far as this reviewer is concerned, there is still a doubt about the use of artificial or political delimitations for biogeographical approaches, although this approach is no longer within the remit of the review of the manuscript and the clarifications made by the authors are sufficient. 

Response: We sincerely thank you for underlining this deficiency, Your suggestion will be a good reference for us to study the plant diversity in Gansu in the future, which will help to improve our research. This also puts forward better requirements for our future collection of distributed data.

Q2: On the other hand, the authors still do not clarify the formula for line 293, the authors refer to Baselga (2012), however, comparing both formulas, it can be seen that they call Sorensen's dissimilarity index βsim=1-(a/(min(b,c)+?)) while Baselga 2012, in table 1, calls Sorensen's index the following formula: (b+c)/(a+b+c). The authors, in line 293, call Sorensen's Dissimilarity a formula that Baselga (2012) calls Simpson's index (more precisely, it is a rotation of Sorensen's index). This reviewer sees the use of this index as appropriate and has no objection to it, but the authors contradict themselves. According to Baselga (2012), the formula given in line 293 belongs to the "Sorensen family of indices" but does not correspond to the Sorensen index. Baselga (2012) in table 1, specifies: 

Sorensen: βsor= (b+c)/(2a+b+c)

Simpson: βsim = min (b,c)/(a+min(b+c))

(this last formula corresponds to the one shown on line 293).

Response: We sincerely thank the reviewer for careful reading.

(Table from Baselga. 2012)

In Baselga (2012), it is indicated in Table 1 that "Simpson dissimilarity (= turnover component of Sørensen dissimilarity)". Initially, in order to minimize the part of overlap with existing articles in our manuscript, we employed the term "turnover component of Sørensen dissimilarity" as a replacement for the commonly used "Simpson dissimilarity". We also applied a modification to the original βsim formula, following the approach of Ye et al. (2018), to reduce the part of overlap with existing articles in our manuscript. After transformation, βsim= min (b,c)/(a+min(b+c))=1-(a/(min(b,c)+?)). You are right, to facilitate a more intuitive comprehension for readers of our manuscript, we reverted to employing the original nomenclature and formula in the revised manuscript (Lines 209-220). Thank you for making our manuscript better.  As follow:

To quantitatively evaluate the spatial turnover of floristic composition in Gansu, we utilized the Simpson dissimilarity index (βsim) [64]. The βsim index allows us to assess the taxonomic turnover that occurred between each pair of regions. The calculation of βsim (1) is performed as follows:

                                                                      (1)

Where 'a' represents the count of shared genera between two regions, while 'b' and 'c' correspond to the counts of genera that are unique to each respective region. The values of βsim range from 0 to 1, with 0 indicating an identical generic composition between the regions, and 1 denoting the absence of shared taxa between the regions [64-65]. One notable advantage of the βsim metric is its ability to mitigate the impact of species richness heterogeneity across different regions, thus ensuring a robust assessment of compositional turnover [64].
